# Multi-layered molecular profiling informs the diagnosis and targeted therapy of desmoplastic small round cell tumor

Desmoplastic small round cell tumor (DSRCT) is an ultra-rare sarcoma with limited treatment options. Here, we show that comprehensive molecular profiling informs diagnosis and individualized therapy in this disease. We report the results of whole-genome/exome, transcriptome, and DNA methylome analyses performed in 30 refractory DSRCT patients, complemented by (phospho)proteomic profiling in nine, within a nationwide precision oncology program. In eight patients (27%), DSRCT was diagnosed only after molecular profiling. Although DSRCTs have "quiet" genomes, 28 patients (93%) received 107 molecular-based management recommendations, including assessment of clinical trial eligibility in 17 (57%). Most recommendations are informed by overexpression of tyrosine kinases, SSTR3/5, and CLDN6, detected in 45%, 33%, and 20% of cases, respectively. Thirteen patients (46%) received recommended therapies, yielding disease control in eight (62%), including three long-lasting responses to pazopanib and trastuzumab deruxtecan, the latter administered based on ERBB2 overexpression in the absence of aberrant ERBB2 kinase activation. These findings demonstrate that multi-omics profiling provides clinically actionable insights for DSRCT management.

Desmoplastic small round cell tumor (DSRCT) is an ultra-rare, high-grade soft-tissue sarcoma (incidence, 0.2/1,000,000 persons/year) of uncertain cellular origin, predominantly affecting male children and young adults[1,2]. It is characterized by a pathognomonic chromosomal translocation, $t(11;22)(p13;q12)$[3,4], resulting in a chimeric protein that contains the N-terminal domain of Ewing sarcoma breakpoint region 1 (EWSR1) and three of four zinc finger domains of Wilms tumor 1 (WT1)[5,6] and is essential for the viability and proliferation of DSRCT cells in preclinical models[7]. The diagnosis of DSRCT can be challenging due to ambiguous histology and expression of neuroendocrine markers and/or cytokeratins[8].

The clinical management of DSRCT involves a multimodal approach, including chemotherapy, surgery, and radiotherapy. Complete or cytoreductive surgery is the most critical component, positively impacting overall survival[9,10]. However, DSRCT is typically diagnosed at an advanced stage, and the overall prognosis is poor, with most patients succumbing to the disease within three years of diagnosis[11]. Due to the lack of clinical trials for this orphan disease, no standard systemic therapy has been established. To date, the use of up to 30 different chemotherapy protocols has been reported, and most patients are treated with regimens adopted from Ewing sarcoma or other soft-tissue sarcomas[9,12].

Previous molecular studies have shown that DSRCT has a low mutational burden and few druggable targets[11,13–15]. In addition, the tumors are immune-cold, with little benefit from immunotherapies, although individual responses have been reported[13,16]. Various multi-targeted tyrosine kinase inhibitors (TKIs), used without molecular biomarker guidance, have shown limited efficacy in individual patients and small case series[17–19]. Retrospective studies suggest some activity of pazopanib, although the determinants of clinical benefit remain unclear[20,21]. Overall, there is an urgent need for effective DSRCT drugs, and it seems warranted to explore whether more comprehensive biological profiling could uncover new therapeutic targets.

✉ e-mail: stefan.froehling@nct-heidelberg.de

A rapidly expanding area of drug development focuses on therapies targeting specific antigens on tumor cells, independent of mutations or dependence on associated signaling pathways. Key examples include chimeric antigen receptor (CAR) T cells, bispecific antibodies, and next-generation antibody-drug conjugates (ADCs). These modalities have considerably broadened treatment options for previously hard-to-target cancers. However, rare malignancies are understudied with respect to these drugs and have therefore benefited less than more common cancers. In DSRCT, in vitro studies have postulated that the signaling pathway controlled by the ERBB (also called HER) family of receptor tyrosine kinases (RTKs) might be activated, and very high doses of the EGFR (also called ERBB1)-directed antibody cetuximab and the pan-ERBB small-molecule inhibitor afatinib, as well as the ERBB2 (also called HER2)-directed ADC trastuzumab deruxtecan (T-DXd), reduced tumor growth in mouse xenografts[22,23]. Based on these findings and given the efficacy of T-DXd in breast cancer, even with (ultra-)low ERBB2 expression[24], and other epithelial malignancies[25], ERBB2 emerges as a promising target in DSRCT. However, responses to ADCs and their correlation with target expression and activation remain largely unexplored in DSRCT patients. A contemporaneous case series by Brahmi et al.[26] reported three DSRCT patients treated with T-DXd based on elevated *ERBB2* mRNA expression, with follow-up comprising three, five, and six treatment cycles, respectively. ERBB2 activation status was not analyzed, leaving open the question of whether *ERBB2* overexpression in DSRCT reflects functional signaling activity or a therapeutically targetable surface expression state.

The MASTER (Molecularly Aided Stratification for Tumor Eradication Research) program (ClinicalTrials.gov: NCT05852522), a prospective observational study conducted by the German Cancer Research Center (DKFZ), the National Center for Tumor Diseases (NCT), and the German Cancer Consortium (DKTK), leverages whole-genome/exome sequencing (WGS/WES), RNA sequencing (RNA-seq), DNA methylation profiling, proteomics, and phosphoproteomics to guide treatment in young adults with advanced malignancies and patients with incurable rare cancers[27,28].

In this exploratory study, we present the clinical courses and molecular target landscapes of 30 heavily pretreated patients with advanced DSRCT enrolled in MASTER, aiming to evaluate the clinical utility of comprehensive multi-omics profiling in this rare and difficult-to-treat cancer. In eight patients, molecular characterization corrected the initial diagnosis to DSRCT. Furthermore, among other treatment outcomes, we describe durable responses to T-DXd lasting up to 24+ months, despite the absence of aberrantly increased ERBB2 activation. Together, our findings demonstrate how multi-layered molecular diagnostics beyond the current standard of care can inform the clinical management of DSRCT patients and lay the groundwork for biomarker-guided clinical trials.

## Results

### Patient characteristics and previous treatments

Between 2013 and 2022, 30 DSRCT patients underwent multi-layered molecular profiling within the DKFZ/NCT/DKTK MASTER program, which enrolls adults younger than 51 years with advanced malignancies as well as patients of any age with advanced rare cancers, and were therefore included in this study. Their median age at the time of molecular analysis was 30 years (range, 18–56); four patients (13%) were female and 26 (87%) male. The median interval between cancer diagnosis and molecular analysis was nine months (range, 1–218). Median survival from the first inter-institutional molecular tumor board (MTB) based on these analyses was 2.1 years (95% confidence interval [CI], 1.0–2.6 years), with a four-year survival rate of 10% (95% CI, 2.8–37%). The median follow-up duration was 17 months (range, 0–48).

Patient characteristics are detailed in Table 1. In eight patients (27%), the initial diagnosis was incorrect or incomplete (Fig. 1a): Three

### Table 1 | Clinical characteristics of 30 DSRCT patients

| Age at enrollment (years) | |
| --- | --- |
| Median | 30 |
| Range | 18–56 |
| **Age at diagnosis (years)** | |
| Median | 27 |
| Range | 18–55 |
| **Sex** | |
| Male | *n* = 26 (86.7%) |
| Female | *n* = 4 (13.3%) |
| **Primary tumor site** | |
| Intra-abdominal | *n* = 27 (90%) |
| Extra-abdominal | *n* = 3 (10%) |
| **Metastatic disease at diagnosis** | |
| Yes | *n* = 23 (76.7%) |
| No | *n* = 7 (23.3%) |
| **Metastatic site at diagnosis** | |
| Lymph nodes | *n* = 18 (60%) |
| Liver | *n* = 10 (33.3%) |
| Lung | *n* = 7 (23.3%) |
| Bone | *n* = 5 (16.7%) |

tumors were classified as carcinoma of unknown primary site (CUP), one as neuroendocrine carcinoma, and one as angiomatoid fibrous histiocytoma; three patients received incomplete sarcoma diagnoses (Ewing sarcoma-like sarcoma, undifferentiated sarcoma with myogenic differentiation [rhabdomyosarcoma-like], and undifferentiated anaplastic sarcoma). In all eight patients, the diagnosis of DSRCT was supported by detection of an *EWSR1::WT1* fusion, further corroborated by gene expression data and a DNA methylation-based sarcoma classifier[29]. In five patients with available tissue samples (DSRCT-04, DSRCT-08, DSRCT-10, DSRCT-14, and DSRCT-20), histopathologic re-evaluation validated the diagnosis of DSRCT. The median time from initial to DSRCT diagnosis was 10.5 months (range, 1–224 months). Among the 27 patients in whom the *EWSR1::WT1* fusion was detected by RNA-seq, 25 (93%) had breakpoints in exons 7–8, one in exons 9–8, and one in exons 10-8 (Supplementary Fig. 1).

Before enrollment in MASTER, patients had received a median of three (range, 1–10) lines of systemic treatment and a median of four (range, 1–12) lines of local therapy, including surgery, radiation, and hyperthermic intra-peritoneal chemotherapy (HIPEC). In addition to systemic therapy, 17 patients (57%) had undergone at least one surgical procedure, with six (20%) also receiving radiotherapy and five (17%) undergoing HIPEC. One patient (3%) had received systemic treatment, radiotherapy, surgery, and HIPEC (Fig. 1b, c and Supplementary Dataset 1). Due to the lack of an evidence-based standard, systemic treatments were heterogeneous. Eighteen patients (60%) had been treated with the VIDE regimen (vincristine, ifosfamide, doxorubicin, etoposide) established for Ewing sarcoma. Six patients (20%) had received targeted therapy as an individual approach. Treatment details and outcomes are summarized in Fig. 1c–e.

### Clinical decision-making based on multi-layered molecular profiling

**Overview.** For clinical decision-making by the inter-institutional MTB of the MASTER program[28,30,31], molecular biomarkers identified through multi-omics profiling (Supplementary Fig. 2a) and resulting clinical management recommendations were grouped into nine intervention baskets: tyrosine kinase (TK), DNA damage repair (DDR), immunotherapy (IT), PI3K-AKT-mTOR (PAM), RAF-MEK-ERK (RME), cell cycle (CC), theranostics (THER), antibody-drug conjugate (ADC),

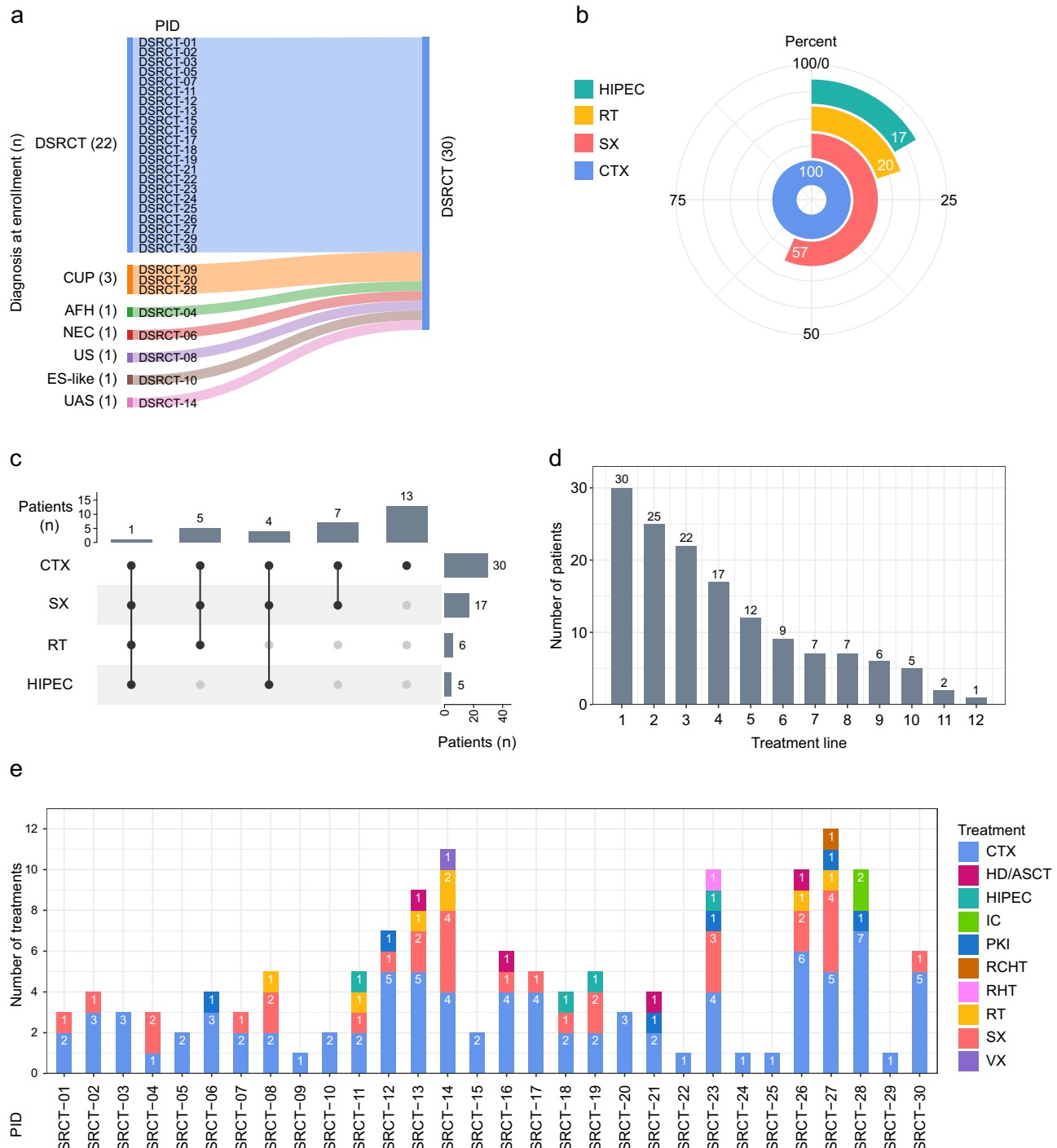

**Fig. 1 | Patient characteristics and previous treatments. a** Initial diagnoses of 30 DSRCT patients analyzed in MASTER. Eight patients were recommended for diagnostic re-evaluation due to the detection of an *EWSR1::WT1* fusion. **b** Proportion of patients receiving the indicated therapies before enrollment in MASTER. **c** Number of patients receiving various therapies before enrollment in MASTER. **d** Number of patients per treatment line. **e** Patient-level representation of therapies administered before enrollment in MASTER. AFH, angiomatoid fibrous histiocytoma; CTX, chemotherapy; ES, Ewing sarcoma; IC, immune checkpoint inhibition; HD/ASCT, high-dose chemotherapy/autologous stem cell transplantation; HIPEC, hyperthermic intra-peritoneal chemotherapy; NEC, neuroendocrine carcinoma; PID, patient identifier; PKI, protein kinase inhibition; RCHT, radiochemotherapy; RHT, regional hyperthermia therapy; RT, radiation therapy; SX, surgery; UAS, undifferentiated anaplastic sarcoma; US, undifferentiated sarcoma; VX, vaccination.

and other (OTH). The MTB provided at least one treatment recommendation (range, 1–7) in 28 of 30 patients (93.3%); in the remaining two cases (DSRCT-02 and DSRCT-08), no actionable biomarkers were identified (Fig. 2a, b). In six patients, sequential samples (up to three) were analyzed. For five of these (DSRCT-07, DSRCT-10, DSRCT-13, DSRCT-24, and DSRCT-30), a second MTB was held, whereas patient DSRCT-21 died before a second MTB could be convened. Two patients had a third MTB. For patient DSRCT-30, no recommendation was issued after the first MTB due to quality limitations of the molecular data. Of the total of 107 recommendations, 48 (45%) fell into the TK basket, followed by the DDR ($n = 13$, 12%), IT ($n = 12$, 11%), and THER ($n = 11$, 10%) categories (Fig. 2a, b and Supplementary Dataset 2,

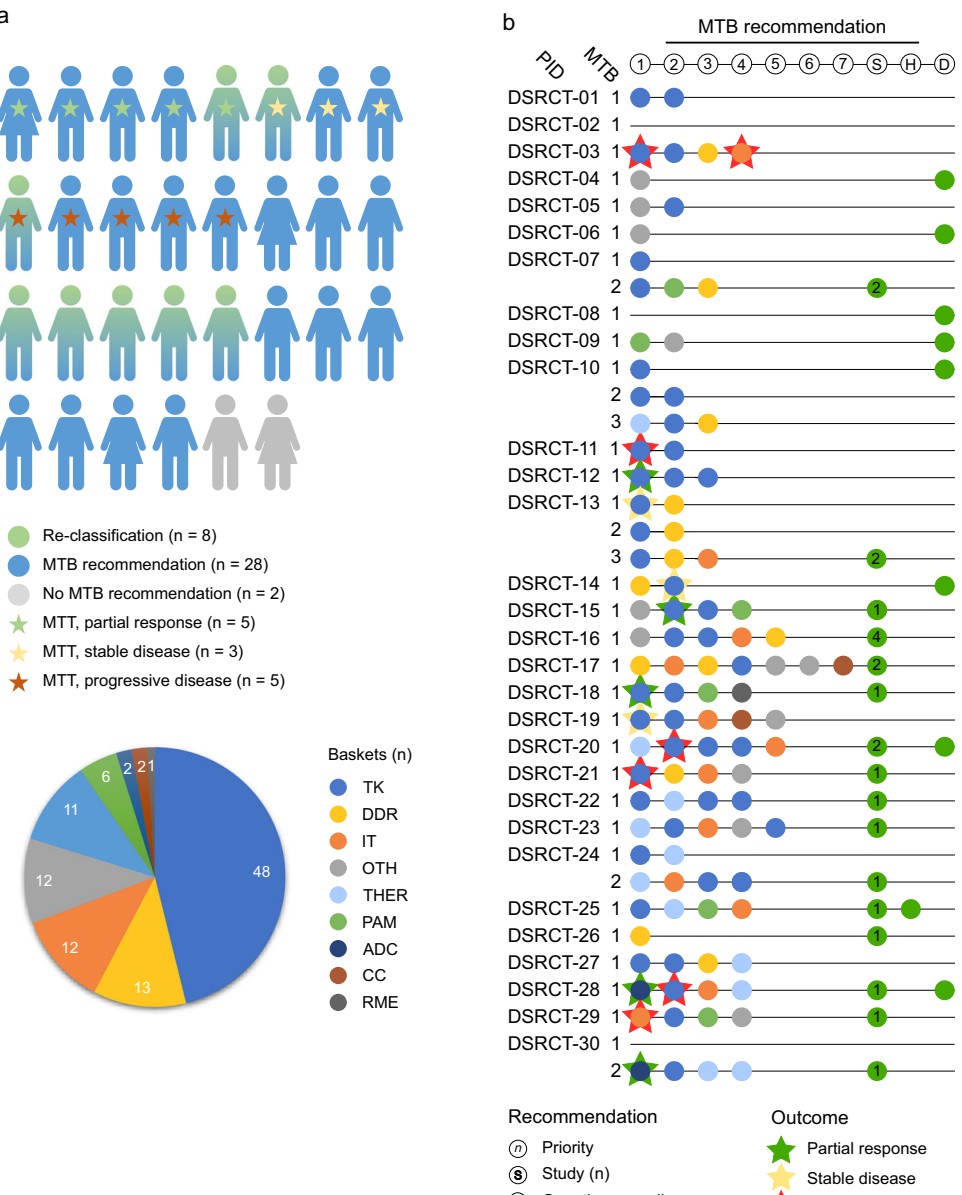

**Fig. 2 | Clinical decision-making and personalized therapy based on molecular profiling. a** Overview of MTB recommendations, molecularly targeted treatment (MTT) recommended by the MTB outside of clinical trials, and patient outcomes. Twenty-eight of 30 patients (93.3%) received one or more treatment recommendations, whose percent distribution across nine intervention baskets is shown in the pie chart. Details on the intervention baskets are provided in Supplementary Table 1. Patients in whom DSRCT had been diagnosed as part of a re-evaluation prompted by molecular profiling (*n* = 8) are indicated in green. **b** Detailed representation of MTB recommendations, MTT, and patient outcomes for implemented therapies (green, yellow, or red star). In patient DSRCT-30, the first MTB recommended re-analysis due to low tumor cell content. Recommendations for enrollment in a clinical study (S), for counseling by human genetics (H), and for diagnostic re-evaluation based on molecular findings indicating DSRCT (D) are shown next to the number of MTB recommendations. PID, patient identifier.

Supplementary Table 1). In five patients, sequential analysis of additional tumor samples showed increasing somatic mutation rates (Supplementary Fig. 2b). Seventeen patients (57%) were recommended for enrollment in 14 different clinical trials (Fig. 2b and Supplementary Table 2)[30].

**DNA analysis identifies few therapeutically actionable targets.** Consistent with previous studies[13,15,32], WGS/WES revealed few somatic mutations (median of 0.68 single-nucleotide variants [SNVs] and small insertions/deletions [indels] per megabase and median of 23 non-silent SNVs and indels per sample). Two genes showed acquired mutations in three patients (10%; Supplementary Fig. 3): *DCC*, encoding the netrin 1 receptor implicated in axon guidance and various epithelial cancers[33,34], which was exclusively altered in female patients, and *EPB41L3*, encoding a cytoskeletal component linked to tumor and/or metastasis suppression[35]. Of a total of 107 treatment recommendations, only four (4%; Supplementary Table 3) were based on individual SNVs. In patient DSRCT-01, missense mutations in *FLT1* and *EPHA3* led us to recommend a multi-targeted TKI, e.g., pazopanib or dasatinib. In patient DSRCT-05, a missense mutation in *FGFR4* provided a rationale for a multi-targeted TKI, e.g., pazopanib. In patient DSRCT-09, a likely gain-of-function mutation in the kinase domain of *MTOR* prompted the recommendation of an mTORC1 inhibitor, e.g., everolimus.

In addition to individual SNVs, eleven of the 107 treatment recommendations were made based on single-base substitution (SBS) signatures (10%; Supplementary Table 4). Detection of signatures SBS3

and/or SBS8, indicating deficiencies in the homologous recombination (HR) and nucleotide excision DNA repair pathways, contributed to the recommendation of a poly(ADP-ribose) polymerase (PARP) inhibitor in 10 patients. In some cases, these signatures co-occurred with alterations of HR-related genes or high expression of *SLFN11*, a potent biomarker of sensitivity to DNA-damaging agents across cancer types, owing to its ability to bind single-stranded DNA and block replication under stress[36,37]. Three of these patients were recommended for enrollment in a clinical trial (NCT03127215) investigating the combination of the PARP inhibitor olaparib and trabectedin in HR-deficient cancers[38].

The overall landscape of somatic DNA copy number aberrations observed in our cohort (Supplementary Fig. 4) was largely consistent with previous studies[15,32,39]. Specific copy number variants (CNVs) were the basis for nine of the 107 treatment recommendations (8%; Supplementary Table 5). For example, deletion and low mRNA expression of *PTEN* in two patients led us to recommend an mTORC1 inhibitor. Additionally, three patients were recommended treatment with a PARP inhibitor due to deletions of genes involved in HR-mediated DNA repair.

The systematic evaluation of rare germline alterations in 101 cancer predisposition genes (Supplementary Table 6) identified one likely pathogenic variant: patient DSRCT-25 had a heterozygous *SDHC* p.R133X stopgain variant associated with hereditary pheochromocytoma and paraganglioma[40]. However, there was no evidence that this variant was relevant for DSRCT development, and no paragangliomas were reported in the patient or the family with a diverse tumor spectrum.

Finally, uniform manifold approximation and projection (UMAP) analysis of the DNA methylation profiles of 30 samples from 25 DSRCT patients, combined with previously published profiles of 305 other sarcomas, including small blue round cell tumors with *BCOR* or *CIC* alterations[29,41], showed that the DSRCT samples formed a distinct and coherent cluster (Supplementary Fig. 5a). All but one DSRCT sample had a sarcoma classifier score > 0.9 (Supplementary Fig. 5b), underlining the diagnostic utility of epigenomic analysis in this entity.

Together, these data demonstrated that exhaustive DNA-based analyses enhance diagnostic accuracy in DSRCT and may identify occasional patients with pathogenic germline variants but are of limited value for identifying new therapeutic targets.

**RNA analysis yields several recurrent therapeutic targets.** Seventy-eight of the 107 recommendations by the MTB (73%) were based on increased mRNA expression of potential therapeutic targets, most of which fell into the TK, DDR, IT, and OTH baskets. The largest proportion of expression-based recommendations (48 of 107, 45%) was accounted for by genes encoding components of TK pathways that can be targeted with clinically approved small-molecule inhibitors (Fig. 3a). As described below, all of the implemented therapies were informed by RNA-based biomarkers. In 10 of 30 cases (33%), the MTB recommended somatostatin receptor 3 (SSTR3)-targeted peptide receptor radionuclide therapy (PRRT). This was based on the consistent overexpression of *SSTR3* and *SSTR5* but not *SSTR1*, *SSTR2*, and *SSTR4* mRNA in DSRCT compared to other sarcomas enrolled in MASTER (Fig. 3b; Supplementary Fig. 6). However, while one patient underwent DOTA-PTR-58 imaging to verify SSTR3 expression and tracer uptake, none received PRRT. In four of 30 patients (13%), androgen receptor (AR) blockade was recommended due to high *AR* expression. In addition, we observed extreme expression of *CLDN6*, encoding a cell adhesion molecule, in six of 30 patients (20%; Fig. 3b), which prompted the recommendation to consider enrollment in a clinical trial of CLDN6-specific CAR T cells (NCT04503278)[42]. This recommendation was implemented in two patients (DSRCT-13 and DSRCT-29). Finally, the MTB recommended ERBB2-targeted treatment in seven patients due to increased *ERBB2* mRNA expression (Fig. 3b), which in two cases led

to the administration of T-DXd, as described below. Besides these recurring treatment recommendations, increased *SLFN11* expression in a patient who also had signature SBS8 prompted the MTB to recommend a PARP inhibitor (Supplementary Table 4), which was administered in combination with trabectedin analogous to, but outside of, the clinical trial mentioned above (NCT03127215), resulting in disease stabilization (Fig. 5).

**Proteome and phosphoproteome analysis provides non-redundant information on pathway activation.** We recently integrated proteomic and phosphoproteomic profiling into the clinical workflow of the MASTER program[43]. Using this newly established pipeline, we retrospectively analyzed samples from nine DSRCT patients for whom suitable tumor tissue was available. For comparison, we analyzed a heterogeneous group of 554 samples representing over 20 sarcoma subtypes (Fig. 4a). UMAP analysis of the global expression of approximately 4000 proteins showed that the DSRCT samples formed a distinct cluster (Fig. 4b). In differential expression analysis, high levels of ERBB2 and CLDN6 were detected in all and two of nine DSRCT patients, respectively (Fig. 4c–e). The correlation of ERBB2 mRNA and protein expression was moderate in both the DSRCT samples and the comparison cohort ($r = 0.5$ and $r = 0.73$, respectively; Fig. 4f). Analysis of the expression and phosphorylation landscape of 43 RTKs showed that, in addition to ERBB2, TYRO3 was overexpressed in most tumors, while high levels of KDR (also known as VEGFR2), MERTK, ALK, and INSR were detected in one or two patients (Fig. 4g). Phosphoproteomic analysis revealed aberrant activity (indicated by a phosphoprotein score > 2) of FGFR4 in patient DSRCT-10 and of KDR and TYRO3 in patient DSRCT-18. In contrast, although ERBB2 protein was consistently expressed, phosphoproteomic analysis revealed no evidence of aberrant ERBB2 kinase activation (Fig. 4g). This distinguishes target expression from driver activation and underpins the therapeutic prioritization of ERBB2-directed ADCs over kinase-dependent inhibitors in DSRCT patients.

**Implementation of molecularly guided treatment recommendations**

**Overview.** Of the 107 targeted therapies recommended, 16 (15%) could be administered in 13 of 30 patients (43%; Figs. 2a, b and 5). All were based on target gene RNA expression, complemented in one case (DSRCT-14) by WES-based detection of signature SBS8. Disease control was achieved in eight of 13 patients (62%; partial remission [PR], $n = 5$; stable disease [SD], $n = 3$). In addition, three patients received chemotherapy according to the VIDE regimen based on the detection of an *EWSR1::WT1* fusion and histopathologic re-evaluation.

**Tyrosine kinase inhibition.** Ten patients received a multi-targeted small-molecule TKI (Fig. 2a, b; Fig. 5). Specifically, pazopanib was administered in nine patients due to overexpression of various combinations of *FGFR2*, *FGFR4*, *FLT4*, *FYN*, *KDR*, *LCK*, *MERTK*, *NTRK3*, *PDGFA*, *PDGFRB*, and *VEGFA*; in one of these cases, overexpression of *NRAS*, *BRAF*, *RAF1*, and *MEK2* provided further support, as it has been postulated that pazopanib also acts as a pan-RAF inhibitor[44]. Five patients (56%) achieved disease control (PR, $n = 3$; SD, $n = 2$), and four (44%) had disease progression. Of particular note is patient DSRCT-18, who showed a PR lasting 17 months and was lost to follow-up on pazopanib therapy. In addition to high *KDR*, *FYN*, and *PDGFRB* mRNA expression, the recommendation of pazopanib in this patient was retrospectively supported by phosphoproteome analysis, which showed increased activity of KDR and TYRO3 (Fig. 4g). One patient (DSRCT-11) whose tumor overexpressed *NRG1* received the pan-ERBB inhibitor afatinib, which has activity in *NRG1*-rearranged neoplasms[45]. However, treatment was discontinued after one month due to generalized disease progression with ascites and colitis with clinically relevant bleeding.

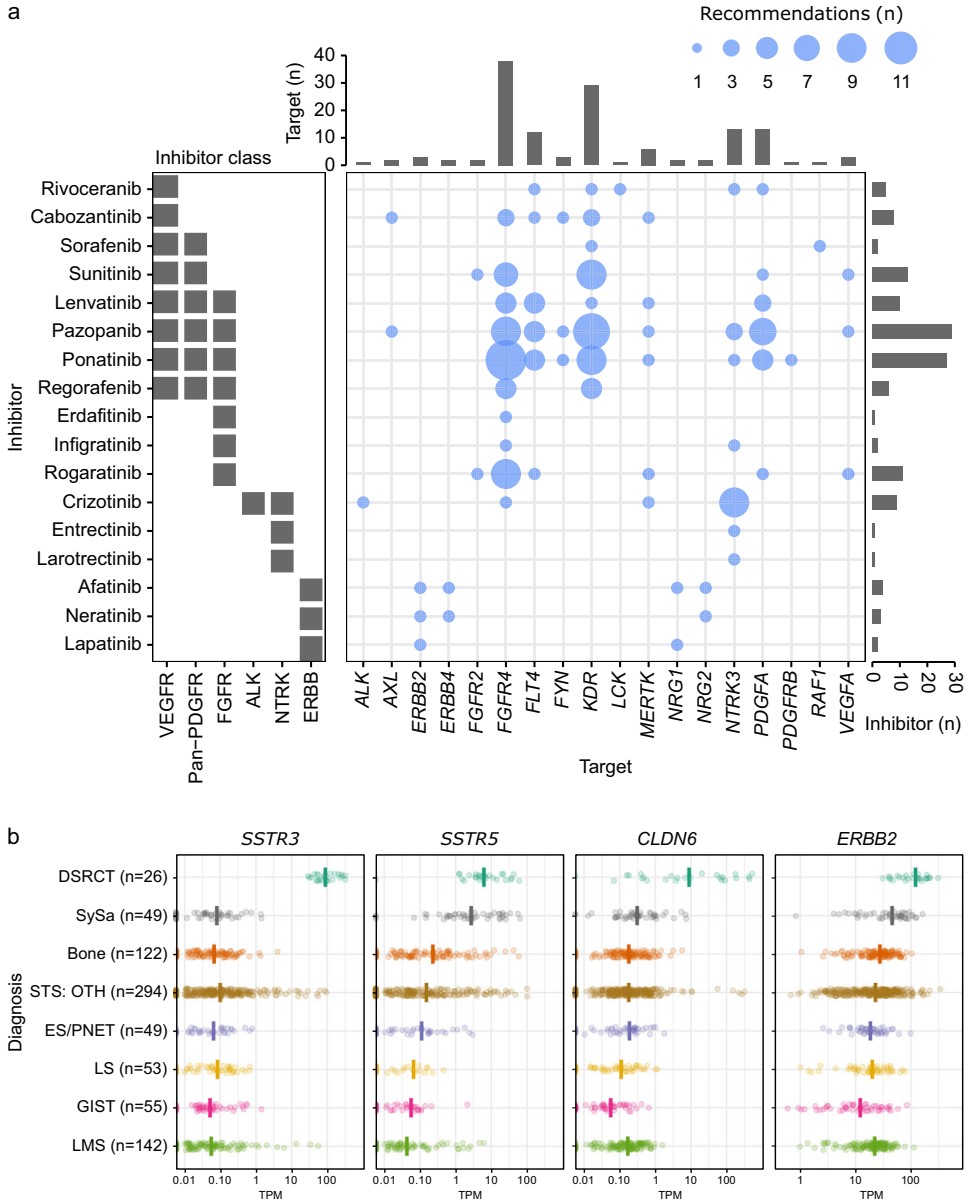

**Fig. 3 | RNA-based clinical decision-making. a** Overview of small-molecule inhibitors recommended by the MTB based on target overexpression. Compounds were assigned to inhibitor classes according to the NCT Drug Precision Oncology Thesaurus Drugs[91]. **b** *SSTR3*, *SSTR5*, *CLDN6*, and *ERBB2* mRNA expression in DSRCT compared to other sarcomas enrolled in MASTER. The vertical lines indicate median expression. GIST, gastrointestinal stromal tumor; LMS, leiomyosarcoma; LS, liposarcoma; OTH, other; PNET, primitive neuroectodermal tumor; SySa, synovial sarcoma; STS, soft-tissue sarcoma; TPM, transcripts per million. Source data for (**a**, **b**) are provided in the Source Data file.

**ERBB2-directed ADC treatment.** Two patients with high ERBB2 mRNA and protein expression levels received off-label treatment with T-DXd (Fig. 2). This selection over other small-molecule- or antibody-based therapies targeting ERBB2 was retrospectively supported by the results of phosphoproteomic profiling, which showed no indication of increased ERBB2 signaling (Fig. 4g). The first patient (DSRCT-28), a 36-year-old man, was diagnosed in February 2018 with poorly differentiated CUP and presented with metastases in the liver, bones, and lymph nodes. At the time of enrollment in MASTER in March 2022, he had undergone 12 lines of therapy, with cisplatin, 5-fluorouracil, and docetaxel, given for four months, and nab-paclitaxel and carboplatin, given for 11 months, each yielding a PR (Fig. 6a and Supplementary Table 7). Molecular analysis revealed an *EWSR1::WT1* fusion, DNA methylation profiling[29] predicted DSRCT with a score of 0.99, RNA-seq confirmed an expression pattern typical of DSRCT, and histologic re-evaluation validated the diagnosis. The MTB provided four treatment

recommendations based on increased target gene expression: (i) small-molecule inhibition of FGFR4, KDR, and MERTK, e.g., with pazopanib, (ii) participation in a clinical trial of CAR T cells against CLDN6 (NCT04503278), (iii) SSTR3-targeted PRRT, and (iv) ERBB2-directed therapy (Fig. 2b and Supplementary Dataset 2). Following the revised diagnosis, the patient was treated with pazopanib but showed disease progression after three months. Next, he received six cycles of chemotherapy according to the VIDE regimen, excluding doxorubicin due to prior disease progression. In April 2023, T-DXd therapy was initiated at 6.4 mg/kg, which was generally well tolerated, with the main adverse effects being grade 2–3 nausea, grade 2–3 loss of appetite, grade 2 fatigue, and grade 1 diarrhea. The patient achieved a partial response at the first staging in July 2023, which was ongoing in July 2024 (Fig. 6a, c and Supplementary Tables 7, 8). Of note, ERBB2 was not detected by routine immunohistochemistry (IHC; Fig. 6c). Given the robust ERRB2 expression identified by mass spectrometry in

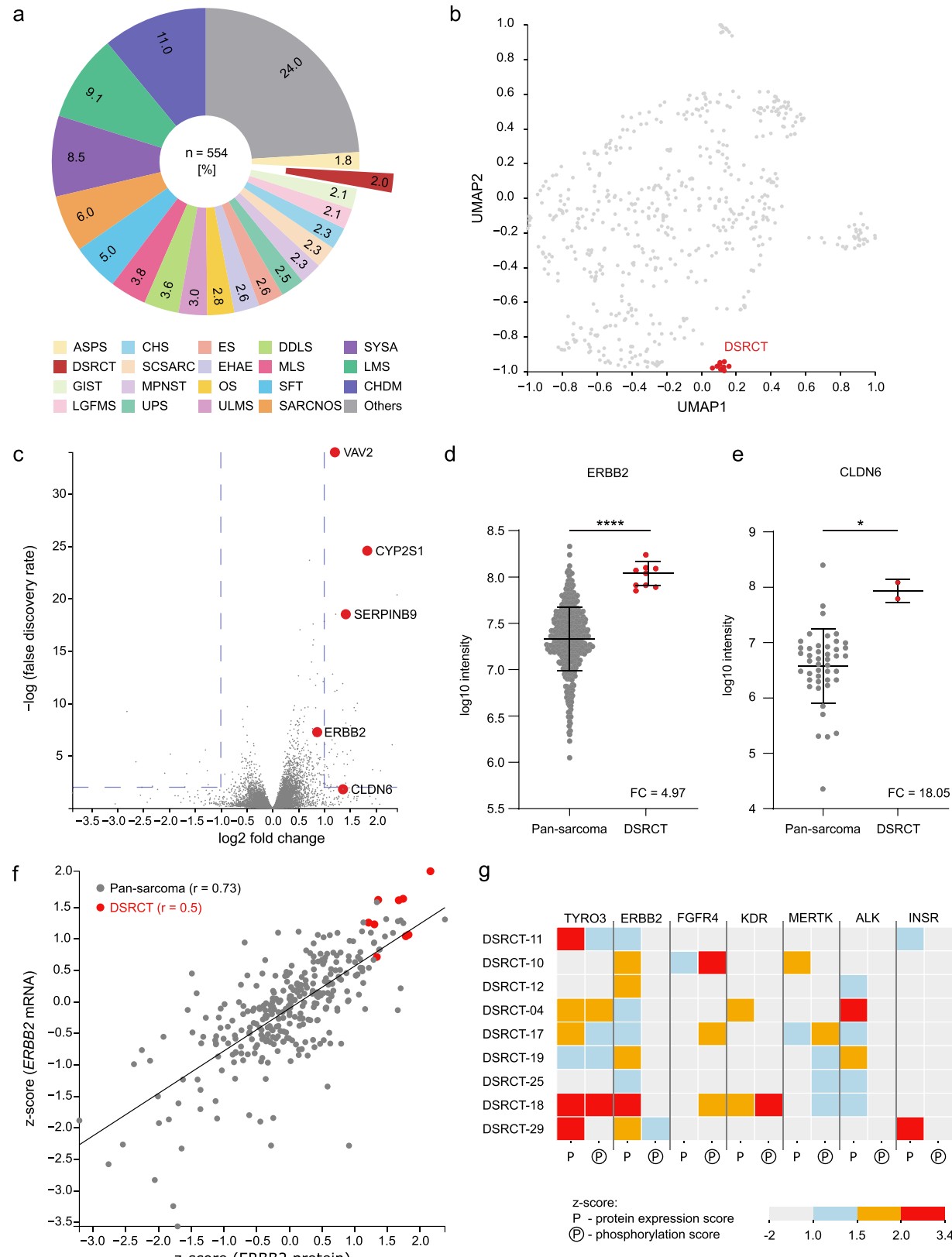

DSRCT patients (Fig. 4d), this discrepancy was likely due to limited quality of the available formalin-fixed and paraffin-embedded (FFPE) tissue.

The second patient (DSRCT-30), a 34-year-old woman, was diagnosed in July 2020 with intra-abdominal DSRCT, suspected peritoneal sarcomatosis, and mesenteric lymph node metastases. Following three

lines of therapy with VAIA (vincristine, adriamycin [doxorubicin], ifosfamide, actinomycin-D), VIDE, and two cycles of trabectedin, she underwent debulking surgery (Fig. 6d and Supplementary Table 9). In October 2022, she was enrolled in MASTER while on sixth-line therapy with irinotecan and temozolomide. Molecular analysis of a peritoneal metastasis resulted in three treatment recommendations: (i) T-DXd,

**Fig. 4 | Proteomic and phosphoproteomic analysis.** (**a**) Composition of the comparison cohort of 554 non-DSRCT sarcoma samples. **b** UMAP analysis of the global expression of approximately 4,000 proteins detected across all patients, showing a distinct DSRCT cluster. **c** Volcano plot summarizing the results of differential protein expression analysis between DSRCT and non-DSRCT sarcoma samples. **d** ERBB2 protein expression in DSRCT (*n* = 9) and the comparison cohort (*n* = 452). **e** CLDN6 protein expression in DSRCT (*n* = 2) and the comparison cohort (*n* = 45). Data are presented as mean ± standard deviation, and statistical significance was determined using Welch's two-tailed *t*-test. *, *p* = 0.0141; ****, *p* = 0.0000000223. **f** Pearson correlation between ERBB2 mRNA and protein expression in DSRCT (red, *r* = 0.5) and the comparison cohort (gray, *r* = 0.73).

**g** Protein expression (P) and phosphorylation (P in a circle) scores of selected RTKs. ASPS, alveolar soft part sarcoma; CHDM, chordoma; CHS, chondrosarcoma; DDLS, dedifferentiated liposarcoma; EHAE, epithelioid hemangioendothelioma; ES, Ewing sarcoma; GIST, gastrointestinal stromal tumor; LGFMS, low-grade fibromyxoid sarcoma; LMS, leiomyosarcoma; MLS, myxoid liposarcoma; MPNST, malignant peripheral nerve sheath tumor; OS, osteosarcoma; SCSARC, spindle cell sarcoma; SARCNOS, sarcoma not otherwise specified; SFT, solitary fibrous tumor; SYSA, synovial sarcoma; ULMS, uterine leiomyosarcoma; UPS, undifferentiated pleomorphic sarcoma; FC, fold change. Source data for (**d**–**f**) are provided in the Source Data file.

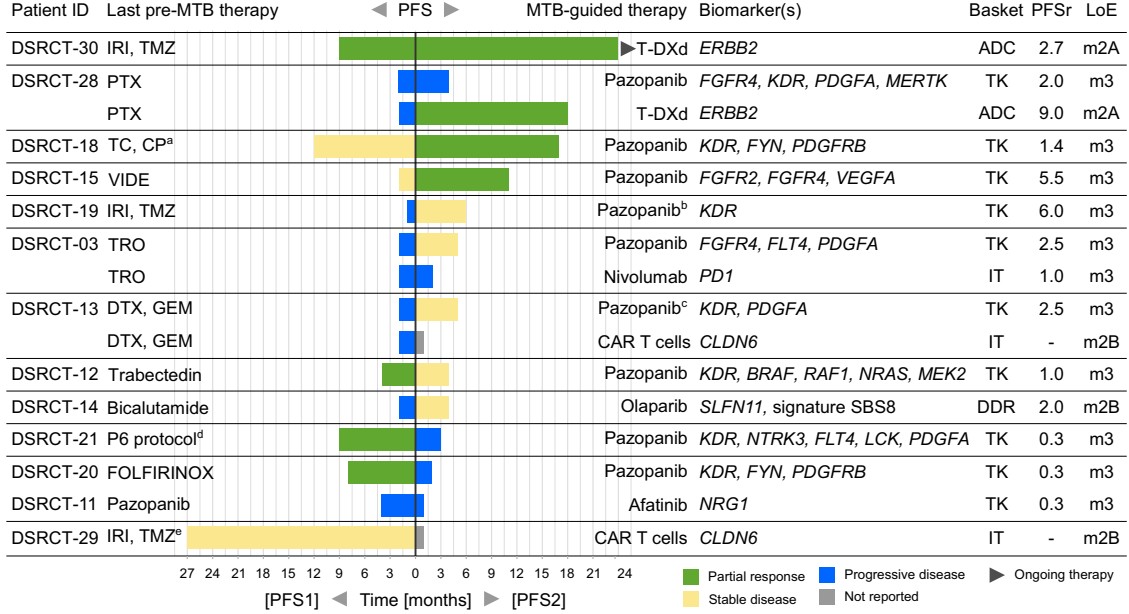

**Fig. 5 | Biomarker-guided therapies implemented.** Comparison of progression-free survival (PFS) associated with the last systemic therapy before MTB evaluation (PFS1) and the subsequent MTB-recommended, biomarker-guided therapy (PFS2), with calculation of the PFS ratio (PFSr = PFS2/PFS1). Clinical benefit was defined as a PFSr ≥1.3. CP, cyclophosphamide; DTX, docetaxel; FOLFIRINOX, folinic acid, fluorouracil, irinotecan, and oxaliplatin; GEM, gemcitabine; IRI, irinotecan; LoE,

level of evidence; PTX, paclitaxel; SBS, single-base substitution; TC, topotecan; TMZ, temozolomide; TRO, trofosfamide; VIDE, vincristine, ifosfamide, doxorubicin, etoposide; [a]in addition to surgery; [b]in addition to gemcitabine; [c]re-exposition; [d]high-dose cyclophosphamide, adriamycin, vincristine (HD-CAV), ifosfamide, etoposide; [e]in addition to surgery and radiation. Source data are provided in the Source Data file.

(ii) trastuzumab, pertuzumab, and atezolizumab within a clinical trial (NCT04551521)[46], both based on increased *ERBB2* expression (Fig. 6f), and (iii) PRRT targeting SSTR3. Upon disease progression, T-DXd was initiated in August 2023. Dosing was reduced to 5.4 mg/kg, 4.4 mg/kg, and 3.2 mg/kg during cycles 3, 5, and 9, respectively, due to grade 2 nausea and grade 3 fatigue. While grade 1–2 fatigue persisted, nausea resolved with dose reduction. After SD in November 2023, the patient achieved a partial metabolic response in January 2024, which was ongoing in September 2024. Following lymphatic progression in May 2025, T-DXd was paused. Interim ⁶⁸Ga-DOTATOC PET/CT imaging was negative. Unexpectedly, restaging in July 2025 revealed a renewed PR, prompting reinitiation of T-DXd (Fig. 6e, h and Supplementary Table 10). In this case, IHC detected moderate membrane expression of ERBB2, corresponding to a score of 2+ (ERBB2-low) according to American Society of Clinical Oncology (ASCO) and College of American Pathologists (CAP) testing guidelines (Fig. 6g). Together, these observations demonstrate the potential of ERBB2-targeted ADC treatment to achieve sustained disease control in DSRCT patients.

## Discussion

In this multi-institutional study, we explored the use of extensive biomarker profiling to inform the clinical management of DSRCT, an aggressive ultra-rare cancer primarily affecting adolescents and young adults that currently lacks established drug treatments, including molecularly targeted approaches. As a hypothesis-generating effort within a prospective precision oncology framework, our analysis highlights how multi-omics investigations, extending beyond the current standard of care, may yield clinically meaningful insights. Specifically, we observed that such integrative approaches can have relevant implications for both precision diagnostics and individualized, mechanism-aware therapy.

More than a quarter of our patients, who were treated at various centers across Germany, initially received incorrect or incomplete diagnoses based on standard histopathology. This reflects the real-world difficulty of classifying small round cell tumors[12] and underscores the potential of molecular methods, such as DNA methylation profiling[29], to improve diagnostic accuracy, independent of the individual examiner's level of expertise with these entities. In all reclassified cases, the finding of an *EWSR1::WT1* fusion was essential, supported by gene expression and DNA methylation profiles characteristic of DSRCT. While we favor an approach in which all sarcomas undergo multi-omics profiling as applied in this study, the diagnostic algorithm in cases consistent with DSRCT according to morphologic criteria should at minimum include a method for detecting this hallmark fusion, such as RNA-seq or fluorescence in situ hybridization. Complementing these approaches, recent work has proposed

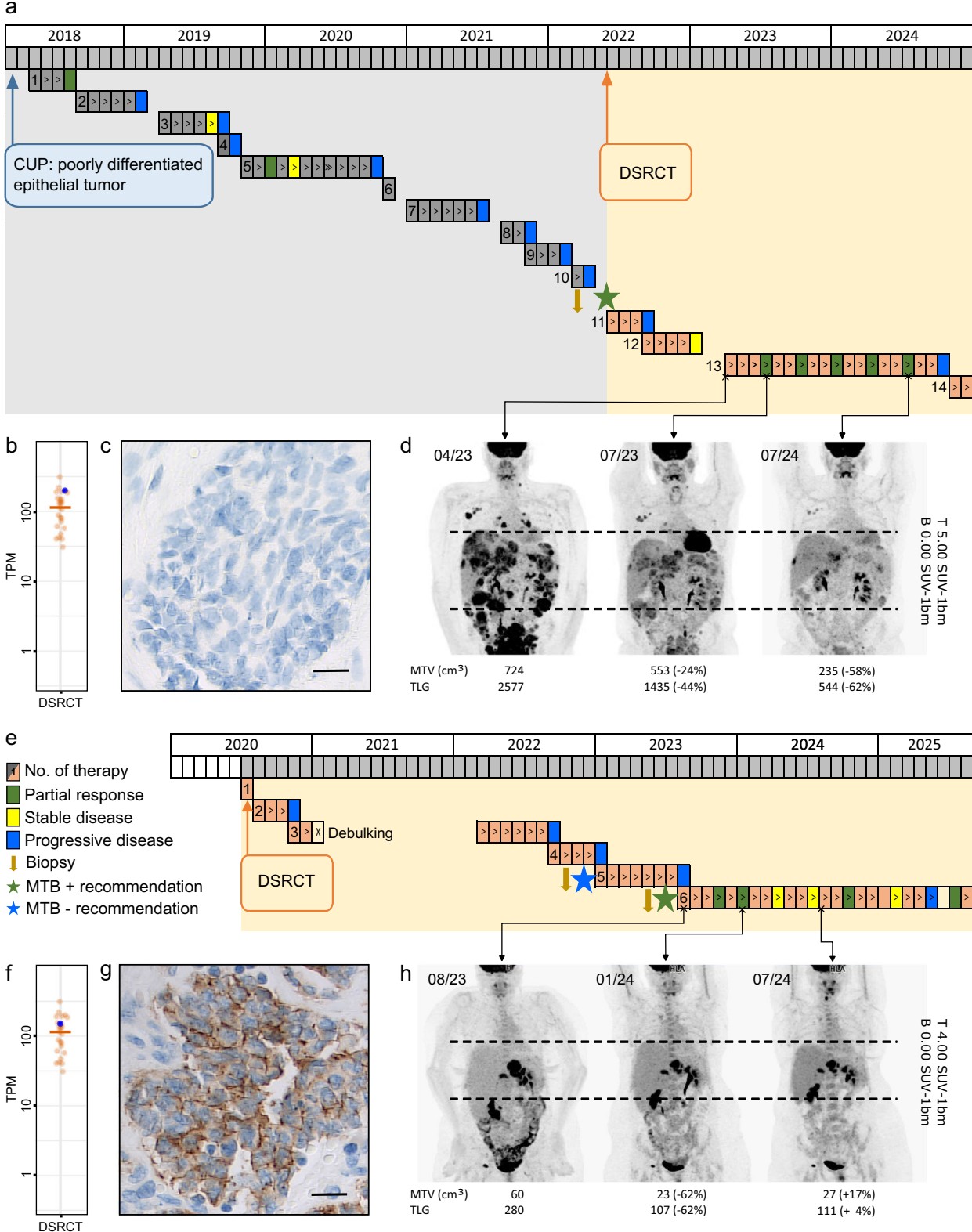

CACNA2D2 as an additional diagnostic biomarker for DSRCT[47]. Taken together, these findings highlight how integrating established and emerging molecular markers can strengthen routine diagnostic workflows and reduce the risk of misclassification in this diagnostically challenging disease.

In the absence of evidence-based drug therapies, an accurate diagnosis of DSRCT currently has limited impact on patient outcomes. However, we believe this will change, and a median time from initial to DSRCT diagnosis of 10.5 months will become unacceptable, as our results underscore in three ways the potential of comprehensive molecular profiling to meaningfully impact therapeutic decision-making. First, the MTB provided recommendations for the vast majority of heavily pretreated patients (93%), demonstrating the feasibility and utility of our approach in a complex clinical setting. Nearly

**Fig. 6 | Efficacy of ERBB2-directed ADC treatment. a** Treatment history of patient DSRCT-28, illustrating 18-month disease control with T-DXd. Details of the individual therapies are provided in Supplementary Table 7. **b** *ERBB2* mRNA expression, determined by RNA-seq, in tumor tissue from patient DSRCT-28. The transcript-per-million (TPM) value of patient DSRCT-28 is indicated by a blue circle, showing its relative position within the cohort (n = 26). **c** ERBB2 protein expression, determined by IHC, in tumor tissue from patient DSRCT-28. **d** Assessment of T-DXd response by positron emission tomography with 2-deoxy-2-[fluorine-18]fluoro-D-glucose integrated with computed tomography ($^{18}$F-FDG-PET/CT) in patient DSRCT-28. The maximum intensity projections of metabolically visible tumor manifestations over time are shown. **e** Treatment history of patient DSRCT-30,

illustrating 24-month disease control with T-DXd. The details of the individual therapies are shown in Supplementary Table 9. **f** *ERBB2* mRNA expression, determined by RNA-seq, in tumor tissue from patient DSRCT-30. The TPM value of patient DSRCT-30 is indicated by a blue circle, showing its relative position within the cohort (n = 26). **g** ERBB2 protein expression, determined by IHC, in tumor tissue from patient DSRCT-30. **h** Assessment of T-DXd response using $^{18}$F-FDG-PET/CT in patient DSRCT-30. The maximum intensity projections of metabolically visible tumor manifestations over time are shown. Scale bars, 20 μm. MPR, metabolic partial response; MSD, metabolic stable disease; MTV, metabolic tumor volume; SUV, standard uptake value; TLG, tumor lesion glycolysis. Source data for (**b**, **c**, **f**, **g**) are provided in the Source Data file.

half of the recommendations (45%) fell into the TK basket, followed by the DDR (12%), IT (11%), and THER (10%) categories, reflecting the diverse multi-omic landscape of DSRCT, a disease usually considered "molecularly silent", and the resulting tailored therapeutic strategies. Second, sequential molecular analyses in a subset of cases highlighted the dynamic nature of DSRCT evolution and prompted additional MTB recommendations. Third, the identification of clinical trial opportunities for 57% of patients underscores the role of molecular profiling in increasing access to innovative therapies.

Different molecular layers contributed variably to clinical decision-making. DNA sequencing confirmed the sparse mutational landscape of DSRCT. Consequently, individual SNVs, such as those in *FLT1*, *EPHA3*, *FGFR4*, and *MTOR*, informed fewer than 4% of treatment recommendations. Similarly, somatic CNVs, e.g., deletions of *PTEN* or HR-related genes, guided MTB recommendations in some cases but were infrequent and often insufficient on their own to dictate therapy. Mutational signatures offered additional value by suggesting vulnerability to PARP inhibitors; however, their utility may be limited by the low overall SNV count in most cases. Finally, a cancer-predisposing germline alteration was detected in only one patient and lacked direct implications for therapy. This aligns with the broader observation that, in contrast to other sarcomas[48], germline findings in established cancer predisposition genes have limited clinical applicability in DSRCT. Larger sample sizes would be needed to analyze possible associations of rare variants in candidate genes or polygenic risk with DSRCT development[49]. Overall, our results suggest that DNA-based analyses, even with comprehensive methods such as WGS/WES, are rarely sufficient for guiding clinical decisions in DSRCT.

Highlighting the need to consider additional molecular layers to broaden therapeutic opportunities for DSRCT patients, RNA-based biomarkers informed 73% of MTB treatment recommendations. A large proportion of expression-based recommendations (45%) were directed at TK pathways, supporting the use of small-molecule inhibitors. Additionally, consistent overexpression of *SSTR3* and *SSTR5* in DSRCT compared to other sarcomas prompted recommendations for PRRT in 10 patients (33%). Although PRRT was not administered in this cohort, our findings laid the molecular groundwork for a recently initiated clinical trial investigating the somatostatin analog pasireotide as maintenance therapy in DSRCT (NCT06456359[50]), which will provide first insights into the value of SSTR-targeted strategies in this disease. Furthermore, extreme *CLDN6* expression in six patients (20%) led to recommendations for enrollment in a clinical trial evaluating CLDN6-specific CAR T cells (NCT04503278)[42]. This strategy illustrates the potential of RNA data to identify novel immunotherapeutic opportunities whose benefit is suggested by the published report of an independent DSRCT patient in this trial in whom SD was achieved by CLDN6-targeted CAR T cells[42]. Similarly, increased *ERBB2* expression prompted recommendations for ERBB2-targeted therapies in seven patients, with two receiving T-DXd. Additionally, RNA data occasionally enhanced DNA-based recommendations. For example, increased *SLFN11* expression in a patient with mutational signature SBS8 supported the administration of a PARP inhibitor combined with trabectedin, resulting in SD. These findings underscore the potential of

RNA-seq for target discovery and refinement of biomarker-driven treatment strategies and support the incorporation of transcriptomic profiling into the management of DSRCT patients. Phosphoproteomic profiling, while available for a smaller subset of patients, provided additional insight by directly assessing kinase activation states. In selected instances, such as the absence of ERBB2 activation in tumors responding to T-DXd and the identification of activated KDR signaling in a pazopanib responder, this would have added critical functional information with the potential to directly influence therapeutic reasoning. These examples illustrate how proteomics can complement other molecular layers and, in specific contexts, guide therapy selection, although inferring kinase activity from phosphoproteomic data remains methodologically challenging[51].

Implementing molecularly guided treatment recommendations was feasible in just over 40% of patients. This reflects challenges common to precision oncology programs, including limited off-label access to targeted agents and a scarcity of molecularly stratified clinical trials – issues that are particularly pronounced in rare cancers like DSRCT[52]. Nonetheless, we successfully acted on several findings, and the disease control rate of 62% in this heavily pretreated population indicates that a rational, biology-guided approach to this disease may confer meaningful clinical benefit. To further quantify therapeutic efficacy, we calculated the intra-patient PFSr, a widely used endpoint in precision oncology[53]. Seven of 13 patients (53.8%) who received MTB-guided therapy achieved a PFSr greater than 1.3, a threshold generally considered indicative of benefit. This compares favorably with landmark precision oncology studies[28,54,55] and exceeds the pooled 38% estimate from a recent meta-analysis[56]. For context, the median time to progression with second-line and third-line chemotherapy in DSRCT has been reported as 3.5 and 2.5 months, respectively[56,57], underscoring the limited efficacy of conventional regimens. While cross-trial comparisons must be interpreted with caution, these data collectively support the clinical relevance of MTB guidance in DSRCT.

Among the 30 patients in our cohort, 10 (33%) received a multi-targeted TKI. In particular, pazopanib was administered in nine patients due to overexpression of various target genes, e.g., *FGFR4*, *FLT4*, *KDR*, *MERTK*, *PDGFA*, *PDGFRB*, and *VEGFA*. Notably, we observed disease control in five of these nine cases (56%), including a PR lasting at least 17 months in patient DSRCT-18. In this case, phosphoproteomic profiling revealed aberrant activation of KDR and TYRO3, kinases within the target spectrum of pazopanib, providing a potential mechanistic explanation for the unusually durable response and illustrating the added value of functional proteomic data in therapy selection.

The efficacy of agents targeting TK pathways has been investigated in several DSRCT case series. Italiano et al. reported on sunitinib treatment of eight patients, which resulted in a PR in two (25%) and SD in three (38%) cases, respectively, and a median progression-free survival (PFS) of 2.6 months[17]. Frezza et al. observed that two (22%) and five (56%) of nine patients treated with pazopanib achieved a PR and SD, respectively, which translated into a median PFS and overall survival (OS) of 9.2 and 15.4 months, respectively[21]. A study by the French registry for the analysis of off-label use of targeted therapies in

sarcomas examined nine patients treated with sunitinib, sorafenib, or bevacizumab. The best response observed was a 5.5-month SD, and the median PFS was 3.1 months[58]. Menegaz et al. reviewed the data of 29 patients treated with pazopanib and observed one complete response (3%), one PR (3%), and 16 cases with SD (55%); the median PFS was 5.6 months, and the median OS was 15.7 months[20]. These converging results demonstrate that TKIs have activity in a subset of DSRCT patients. However, their use was unstratified and relied either on empirical evidence or the general observation that the DSRCT micro-environment is characterized by neoangiogenesis and expression of various angiogenic factors. In contrast, we employed patient-level biomarker profiles to individualize therapy selection. In particular, the long-lasting response to pazopanib in a patient whose transcriptomic and phosphoproteomic profiles revealed aberrant activation of several kinases targeted by this drug underscores the potential of comprehensive, multi-layered molecular analyses. We believe that considering these two diagnostic layers will help maximize the therapeutic potential of pazopanib, and potentially other TKIs, and guide patients unlikely to benefit towards alternative therapies.

Another TK that recently emerged as a potential therapeutic target is ERBB2, whose mRNA levels were consistently higher in DSRCT than in other sarcomas. Expression of ERBB2 has been previously observed in DSRCT patients and preclinical models, suggesting that it may be constitutively activated and can be inhibited by small molecules or antibodies whose efficacy requires dependence on the corresponding signaling pathway[22]. However, when integrating phosphoproteomic analysis with transcriptomic and proteomic profiling, we found no evidence of aberrant ERBB2 kinase activation in DSRCT, despite robust transcript and protein expression. This distinction highlights the difference between targeting ERBB2 based on its surface expression versus inhibiting it as an active signaling driver and argues against the use of agents whose mechanism of action depends on ERBB2 signaling activity or dimerization, such as afatinib, neratinib, or pertuzumab, which have previously been proposed for DSRCT. Instead, it supports the rationale for ADCs like T-DXd, which act independently of receptor activation. Consistent with this, both patients in our cohort who received T-DXd achieved long-lasting remissions, despite having progressed after 12 and five prior therapy lines, respectively. These results provide translational validation, supported by extended patient-level follow-up, of observations in patient-derived xenografts[23] and a contemporaneous case series[26], and indicate that clinical trials of T-DXd, or further-generation ERBB2-targeted ADCs, in this intractable malignancy are warranted.

To translate this signal of ADC efficacy in DSRCT patients into improved routine care, developing robust biomarkers will be essential. Studies in breast cancer, a disease known for its association with ERBB2 expression in a subset of cases, have shown that patients with low or ultra-low ERBB2 levels also benefit from T-DXd[24]. Consequently, the criteria for quantifying this biomarker were quickly adapted, e.g., in the ASCO/CAP testing guidelines. However, the minimal clinically relevant ERBB2 expression remains uncertain, illustrated, e.g., by the DAISY phase 2 trial, in which even patients with ERBB2-negative tumors by standard IHC responded to T-DXd[24]. This might be attributed to T-DXd's strong bystander effect, where the targeted payload release is mediated by ERBB2 expression below the detection limit of IHC, suggesting that alternative methods, such as RNA-seq, may become critical in identifying eligible patients. Of our two long-term responders, both of whom showed readily detectable ERBB2 mRNA levels, one had an ERBB2-low tumor, and one lacked ERBB2 expression by IHC, which, given the robust expression by mass spectrometry, was probably due to insufficient tissue quality. Based on our data, we predict that most DSRCT patients are candidates for ERBB2-targeted ADC therapy and that a negative ERBB2 IHC result does not exclude the possibility of a response to these agents. However, future studies must define the range of clinically relevant target expression and determine the optimal detection methods.

Our study underscores the potential of multi-layered molecular profiling and inter-institutional collaboration for advancing research in rare cancers. While individually infrequent, rare cancers collectively represent a substantial patient population, accounting for 20–25% of all newly diagnosed cancers in many regions worldwide, and are often associated with poor outcomes due to limited treatment options and insufficient research investment[59]. We leveraged the framework of a nationwide precision oncology platform to deliver insights into the clinically actionable molecular landscape of DSRCT, offering a blueprint for accelerating research in other ultra-rare malignancies. Our findings demonstrate, with patient-level evidence and extended follow-up, that multi-omics profiling can refine biomarker interpretation (e.g., distinguishing ERBB2 expression from activation), enable rational selection of targeted agents (e.g., T-DXd and pazopanib), and identify clinically actionable targets in previously understudied disease contexts (e.g., SSTR3/5 and CLDN6 in DSRCT). Together, these results highlight clinical and biological insights that can be gained from multi-layered molecular profiling in DSRCT, providing a foundation for prospective trials and future mechanistic studies.

## Methods

### Patients

The study was approved by the Ethics Committee of Heidelberg University (protocol no. S-206/2011) and conducted in accordance with the Declaration of Helsinki. We retrospectively identified all DSRCT patients enrolled in the DKFZ/NCT/DKTK MASTER program[28] until December 2022. Thirty individuals with advanced disease provided written informed consent for multi-layered molecular profiling of tumor tissue and, in the case of DNA sequencing, a matched blood sample, as well as for the longitudinal collection of clinical follow-up information. Identical consent procedures applied to all patients in the control cohorts. Biological curation and clinical annotation of molecular profiles were performed as previously described, and a multi-institutional MTB involving treating physicians provided recommendations for clinical management[30]. Patients received no compensation for their participation.

### Sample processing

Frozen tissue sections were assessed by board-certified pathologists to determine tumor cell content, including the presence of necrosis. Suitable samples were processed further at the NCT Heidelberg Sample Processing Laboratory. DNA, RNA, and cell lysates from fresh-frozen tumor specimens and DNA from blood samples were obtained using the AllPrep DNA/RNA/miRNA Universal Kit and the QIAamp DNA Mini Kit (Qiagen). Nucleic acids from FFPE samples were extracted using the Allprep DNA/RNA FFPE Kit (Qiagen). Subsequent quality control and quantification steps were performed using a Qubit Fluorometer (Life Technologies) and a 4200 or 2200 TapeStation System (Agilent).

### DNA sequencing

Libraries for WGS were prepared using the Illumina TruSeq Nano DNA Library Prep Kit with 100 ng DNA as input and sequenced on an Illumina HiSeq X Ten or NovaSeq 6000 instrument at the DKFZ Next Generation Sequencing Core Facility. Libraries for WES were prepared using the Agilent SureSelect All Exon Kit v5 or v5 + UTRs with 200 ng DNA as input and sequenced on an Illumina HiSeq 2000, HiSeq 2500, HiSeq 4000, or NovaSeq 6000 instrument.

### Nucleotide sequence alignment

DNA sequencing reads were mapped to an assembly of the human genome version hs37d5 (1000 Genomes Project phase 2) and the Enterobacteria phage phiX174 genome using BWA mem (version

0.7.15) with the −T0 parameter as the only one deviating from the default. BAM files were sorted with bamsort (biobambam package version 0.0.148), and duplicates were marked with markdup (Sambamba package version 0.6.5)[60].

## Somatic SNV and indel calling

Somatic SNVs were detected with an in-house pipeline based on SAMtools (version 0.1.19) mpileup and bcftools and using heuristic filtering as previously described[61–63]. Briefly, initial SNV calls were detected in the tumor BAM file by mpileup, which considered only reads with a minimum mapping quality of 30 (−q30), and bcftools, which reported all positions containing at least one high-quality non-reference base (−vcgN −p2.0), followed by inspection of these positions in the control sample using mpileup. SNVs were annotated with ANNOVAR (version November 2014) using GENCODE (release 19). Downstream filtering discarded variants with low support of the alternative allele, occurring in tandem repeats or other read-attracting regions, with PCR or sequencing strand bias, or with significant bias in the PV4 field of the mpileup output. Somatic SNVs annotated as missense, stopgain, stoploss, or splicing were defined as non-silent. Short indels were detected with Platypus (version 0.8.1.1)[64], and only those that had the filter tag PASS or passed custom filters allowing for low variant frequency were retained. Annotation of indels was performed using ANNOVAR (version February 2016), and calls falling into a coding sequence or splice site were extracted.

## Somatic structural variant calling

Structural variants (SVs) were detected using SOPHIA (version 35)[65], which uses the "supplementary alignment" feature of BWA mem to discover SVs and custom thresholds and a background panel of normals to filter out common variants and recurrent artifacts.

## Somatic CNV calling

DNA copy number profiles of samples subjected to WGS or WES were determined using ACEseq (version 5.0.1) (https://doi.org/10.1101/210807) and CNVkit (version 0.9.3)[66], respectively. For CNVkit data processing, the steps for inferring ploidy and tumor cell content (TCC) were adopted from the ACEseq workflow. In both workflows, segments containing at least 20 heterozygous single-nucleotide polymorphisms (SNPs) were processed to infer sample ploidy and TCC. The algorithm tested every possible combination of TCC (range, 0.15–1.0) and ploidy (range, 1.0–6.5) to find the local minima and, thus, the optimal solution. If more than one solution was possible, they were visually evaluated, and one was selected. The results for patient DSRCT-16 were inconclusive due to the high level of degradation of the matched control sample and therefore removed from further analysis. The remaining 29 samples had an estimated TCC above 30% and were included in the downstream analysis. Genomic gains and losses were annotated when a given genomic segment deviated in ploidy from the overall sample ploidy by + 0.7 or − 0.7, respectively. To determine the frequency of gains and losses across the genome, the copy number segments' breakpoints of all samples were gathered to create non-overlapping genomic sections. For each non-overlapping genomic section, the number of samples with gain, loss, and copy number neutral state was calculated. Segments covered by fewer than 20 patients were discarded, and the frequency of gains and losses was calculated for the remainder. The results were annotated using the Cancer Gene Census (https://cancer.sanger.ac.uk/census).

## Germline variant calling

Germline SNVs and indels were called using Platypus (version 0.8.1.1), germline CNVs were detected using the GATK gCNV module (version 4.2.4.0), and germline SVs were identified using SOPHIA (version 2.2.0). After obtaining raw SNV and indel calls, variants were filtered based on a minimum coverage of 10 x and a minimum support of 3 x

for the alternate alleles. Variants with the Platypus filter tags PASS or alleleBias were considered further. After coverage-based filtering, population-based variant frequencies, i.e., the minor allele frequency from gnomAD (version 2.1) and the variant frequency from an in-house panel of normals comprising 3,910 WGS and 1198 WES samples were added using vcfanno (version 0.3.2)[67]. Variants with a frequency above 0.0005 (minor allele frequency in gnomAD) or 0.5 (variant frequency in normal samples), respectively, were considered common or artifacts and removed from further analysis. Variants in cancer predisposition genes that are annotated in ClinVar as pathogenic or likely pathogenic were excluded from population-based filtering. Additional genomic annotations and variant consensus were added to the filtered variants using VEP (version 104)[68], followed by classification into germline or somatic based on the variant allele frequency obtained from control and tumor samples using TiNDA, which uses an EM-based clustering method in the canopy R package (version 1.3.0)[69]. The GATK gCNV module[70] was used to detect germline CNVs from WGS and WES samples by applying a background cohort model created from 200 WGS or WES samples that were matched with the target capture kit. The workflow adhered to the best practices for detecting genomic CNVs outlined by GATK. For WGS data, one deviation was that only GENCODE (release 19) protein-coding regions were analyzed to expedite the analysis process. Next, CNV segments with a quality control score above 30 were annotated with a subset of gnomAD SV data using vcfanno. CNV segments with an 80% overlap with a common gnomAD SV (minor allele frequency > 0.1%) of the same type were excluded. Additionally, targets within CNV segments were required to have denoised ploidies within the top or bottom 5% of background denoised ploidies in the case of duplication or deletion, respectively. Germline SVs were detected along with somatic SVs using SOPHIA. Finally, germline variants identified in cancer predisposition genes (Supplementary Table 6) were provided to medical geneticists for classification according to American College of Medical Genetics and Genomics/Association of Molecular Pathologists criteria[71] and further ClinGen specifications[72]. Clinically relevant variants were discussed in the MTB and integrated into clinical management recommendations.

## Quantification of tumor mutational burden

The number of mutations per megabase was calculated by dividing the sum of non-silent SNVs and coding indels by the length of the genome's coding sequence in megabases. For WES samples, the denominator was adjusted to the respective target coverage.

## Mutational signature analysis

The presence of SBS signatures from the COSMIC database (version 2)[73] was assessed using the YAPSA R package[74]. The analysis included all somatic SNVs, limited to those within target regions for WES samples. For each sample, a mutation catalog was generated and further corrected for WES samples to account for the different occurrences of triplet motifs within the target sequences, and mutational signatures were computed along with their confidence intervals. Signatures were considered present if they exceeded signature-specific thresholds with a cost factor of 6 (absolute and relative for WGS and WES samples, respectively) and the lower limit of their confidence interval was greater than zero.

## DNA methylation analysis

Tumor DNA (250 ng) was analyzed using Infinium MethylationEPIC BeadChip arrays (Illumina). Raw data were used to run the sarcoma classifier (version 12.2)[29], followed by processing using the preprocessRaw function of the minfi R package[75]. Probes that were unreliable, cross-reactive, mapping to sex chromosomes, overlapping with SNPs, or had a detection $p$ value ≥ 0.01 were filtered out[76,77]. Each sample was normalized using the BMIQ function with default settings from the wateRmelon R package[78]. Beta values were used for further

analyses. Only entities with three or more samples were kept ($n$ = 335). For dimensionality reduction, UMAP analysis was performed using the umap function of the umap R package with default settings based on the algorithm described by McInnes and Healy (10.48550/arXiv.1802.03426). For comparison, raw methylation data of 286 non-DSRCT sarcomas, 18 DSRCT samples (EGAS00001006939), and 19 small blue round cell tumors with *BCOR* or *CIC* alterations (GSE140686) were downloaded[29].

### RNA sequencing and fusion gene detection

Libraries were prepared using the Illumina TruSeq RNA Library Preparation Kit with 1,000 ng total RNA as input or the Illumina TruSeq mRNA Stranded Library Preparation Kit with 500 ng total RNA as input. Three libraries were pooled and sequenced on one lane of an Illumina HiSeq 2500, HiSeq 4000, HiSeq X Ten, or NovaSeq 6000 instrument. Reads were aligned to the same reference genome as DNA sequencing data with STAR 2.5.1b[79]. Fusion transcripts were detected with Arriba (version 2.4.0)[80]. For genes included in the Cancer Gene Census (https://www.sanger.ac.uk/data/cancer-gene-census) as well as in our in-house lists of druggable genes and other genes of interest, RNA overexpression was defined as a fold change of more than two times the median of a comparison cohort of 148 RNA-sequenced samples within the MASTER program. Since October 2017, we have also applied an alternative criterion: for genes never found to be upregulated by a factor of 2, a z-score greater than 2 was considered indicative of overexpression.

### Proteome and phosphoproteome analysis

**Sample preparation.** Immediately after collection, samples were snap-frozen without the addition of medium. Cryopreserved specimens were mounted on cork plates using Tissue-Tek, sectioned into three 50-μm slices, and shipped on dry ice for proteome and phosphoproteome analysis. For lysis, tissue slices were solubilized in 4% SDS and 40 mM Tris-HCl (pH 7.6), followed by ultrasonication and heating (10 min, 95 °C). Residual DNA was subsequently sheared by adding trifluoroacetic acid (2% final concentration) and incubating the lysate for 2 min at 80 °C. The reaction was quenched with 4-methylmorpholine (4% final concentration) to adjust the mixture to neutral pH, after which cell debris was removed by centrifugation (20,000 rpm, 20 min). Protein concentrations were determined by a BCA assay, and samples were stored at −80 °C until further processing. Protein digestion was performed using the SP3 protocol[81]. Briefly, 200 μg of protein per sample was processed. Proteins were precipitated from the lysate onto carboxylate- and amine-modified magnetic beads (Sera-Mag A and B, Sigma-Aldrich) in the presence of 70% ethanol, using a 1:5 protein-to-bead ratio. After reduction and alkylation of cysteine residues, proteins were digested overnight with trypsin (Roche) at a 1:50 (w/w) enzyme-to-substrate ratio. Resulting tryptic peptides were desalted by solid-phase extraction (SPE) using Oasis HLB 96-well plates (Waters). Peptides from each patient sample were then labeled with tandem mass tags (TMT)[82]. Finally, combined TMT-labeled peptides were desalted by SPE using 50 mg tC18 reversed-phase cartridges (Waters). TMT-labeled samples were separated by high-pH reversed-phase HPLC[83], using an X-Bridge BEH130 C18 column (4.6 × 250 mm; Waters) operated at a flow rate of 1 ml/min. A total of 96 fractions were collected at 30-s intervals and subsequently concatenated into 48 fractions for proteome analysis (13% of total material) and 12 fractions for phosphopeptide enrichment (87% of total material). Phosphopeptide enrichment was performed by immobilized metal affinity chromatography using Fe(III)-NTA cartridges on the AssayMAP BRAVO liquid-handling platform.

**Liquid chromatography-tandem mass spectrometry data acquisition and pre-processing.** TMT-labeled tryptic peptide batches were analyzed by liquid chromatography-tandem mass spectrometry using a Vanquish Neo HPLC system coupled to an Orbitrap Eclipse mass spectrometer (Thermo Fisher). For proteome analysis, peptides were separated on an Acclaim PepMap 100 C18 column at a flow rate of 50 μl/min using a 25-min solvent gradient[84]. For phosphoproteome analysis, peptides were separated on a Reprosil Gold column at 300 nl/min using a 90-min solvent gradient[85]. In both workflows, the mass spectrometer was operated in data-dependent acquisition mode. Intact peptide mass spectra (MS1) were acquired in the Orbitrap, and fragment ion spectra (MS2) were generated by collision-induced dissociation (for unmodified peptides) or by multi-stage activation (for phosphopeptides). TMT reporter ion spectra were subsequently acquired using synchronous precursor selection (MS3). Raw mass spectrometry files were processed for peptide identification and quantification using MaxQuant (version 1.6.12.0). Data were searched against the human SwissProt+TrEMBL database (97,057 sequences; downloaded November 2020) supplemented with common contaminants. Searches were conducted using the *MS3 reporter ion* experiment type with *TMT11plex* labels, and isotope impurity correction factors provided by the manufacturer were applied for each TMT lot. Search parameters allowed up to three missed cleavages, a minimum peptide length of six amino acids, a maximum peptide mass of 6000 Da, and a fragment ion tolerance of 0.4 Da (ITMS). The false discovery rate (FDR) was set to 1% at the peptide-spectrum match level and to 100% at the protein level. For phosphoproteomic samples, phosphorylation of serine, threonine, and tyrosine residues was specified as a variable modification. TMT batches were processed with SIMSI-Transfer (version 0.5.0) for missing-data reduction via MS2-level identification transfer[86]. For intensity normalization, MS3 reporter ion intensities were median-centered within each batch, and MS1 intensities were median-centered across batches. Protein grouping, FDR control at 1% for peptide and protein identifications, and MaxLFQ quantification across all TMT batches were performed at the gene level using the *picked protein group FDR* package (version 0.6.2)[87].

**Global differential protein abundance analysis.** For each protein, differences in abundance between DSRCT samples and all other samples in the cohort were assessed using the independent *t*-test (ttest_ind()) implemented in SciPy (version 1.11.4). Multiple testing correction was performed using the Benjamini−Hochberg procedure (fdrcorrection()) implemented in statsmodels (version 0.14.2) to control the FDR. Proteins were considered significantly differentially abundant at FDR < 0.05 and $\log_{10}$(fold change) > 1.

**Single-protein differential abundance analysis.** For ERBB2 and CLDN6, differences $\log_{10}$-normalized protein intensities between DSRCT samples and all other samples in the cohort were assessed using Welch's *t*-test for unequal sample sizes. *p* values < 0.05 were considered statistically significant.

### IHC

IHC was performed on 3 μm tissue sections with the ready-to-use, pre-diluted PATHWAY anti-HER2/neu (4B5) rabbit monoclonal primary antibody (cat. no. 05278368001, lot no. K13629, Ventana/Roche) using a BenchMark ULTRA Autostainer VENTANA (Roche) as previously described[88]. Immunoreactivity was assessed according to ASCO/CAP criteria established for breast cancer[89]. IHC readers were blinded to clinical data.

### Clinical decision-making

Treatment recommendations were formulated by the MTB of the DKFZ/NCT/DKTK MASTER program, an interdisciplinary team of oncologists, pathologists, geneticists, and bioinformaticians. Decisions were based on multi-omics data (WGS/WES, RNA-seq, DNA methylation profiling, proteomic and phosphoproteomic analysis) and integrated multiple parameters, including patient-specific factors (e.g.,

prior therapies, tumor histology, comorbidities) as well as the level of evidence (LoE) supporting the clinical actionability of each molecular alteration (e.g., regulatory approval, clinical trial data, or preclinical evidence). The process followed a standardized, structured workflow supported by extensive manual and automated annotation pipelines, employing a four-tiered system for grading molecular alterations based on clinical and preclinical evidence, as previously described[28,30,31,90]. Briefly, LoE m1a–m2c correspond to clinical evidence ranging from data in the same tumor entity to case reports and small case series in other entities, whereas LoE m3 and m4 were assigned to molecular alterations supported only by preclinical evidence or by mechanistic rationale, respectively. For clinical decision-making, additional considerations such as drug or trial availability and patient-specific factors (e.g., prior treatments, comorbidities) were also taken into account. Clinical follow-up data were collected every three months.

### Reporting summary
Further information on research design is available in the Nature Portfolio Reporting Summary linked to this article.

## Data availability
Sequencing, DNA methylation, and (phospho)proteomic data have been deposited in the European Genome-Phenome Archive (https://www.ebi.ac.uk/ega/datasets) under accession EGAS00001007934. Source data are provided with this paper.

## Code availability
Bioinformatic analyses were performed with the open-source software described above, using parameters defined in the corresponding Methods subsections.

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

## Acknowledgements

We thank the NCT Sample Processing Laboratory and the DKFZ Next Generation Sequencing, Microarray, and Omics IT and Data Management Core Facilities for technical support. This work was supported in part by the German Federal Ministry of Research, Technology and Space (grant 031L0305A to B.K. and grant 01KD2207A to An.S., B.B.-B., S.B., R.F.S., G.M., Al.S., D.B.L., B.K., D.H., H.G., and S.F.), the European Research Council (grant 833710 to B.K.), and the German Research Foundation (grant 452419311 to B.K.). The MASTER program is supported by the NCT Overarching Clinical Translational Trial Program, the NCT Heidelberg Molecular Precision Oncology Program, and DKTK.

## Author contributions

M.R., C.E.H., S.K., H.G., and S.F. conceptualized the study. H.G. and S.F. supervised the study. M.R., C.E.H., C.M., Ca.H., A.B.-M., K.B., L.A., S.B., M.B., C.H.B., D.T.R., T.K., F.K., K.S.-O., R.F.S., G.B., D.B.L., A.J., E.S., Ch.H., M.-V.T., P.H., S.K., H.G., and S.F. collected clinical and molecular data. M.R., M.O., N.P., An.S., J.H., B.H., C.E., A.M., C.B.J., Am.S., M.T., D.B.L., M.S., B.K., and D.H. developed analysis methodology. M.R., M.O., N.P., An.S., J.H., B.H., C.E., A.M., E.K.-H., C.B.J., Am.S., M.T., and P.L. curated clinical and molecular data and designed and performed formal data analyses. M.R., M.O., N.P., and An.S. visualized results. T.P., A.J., and E.S. evaluated germline variants. M.A., G.M., Al.S., and W.H. performed histopathologic analyses. B.B.-B. interpreted radiologic imaging data. M.R., M.O., N.P., An.S., E.K.-H., and S.F. wrote the primary draft, which was reviewed by all co-authors.

## Funding

## Competing interests

M.A. has had an advisory role and received honoraria from AbbVie and Boehringer Ingelheim. G.M. has had an advisory role and received honoraria from Boehringer Ingelheim. D.B.L. has received honoraria from Infectopharm. B.K. is a co-founder and shareholder of OmicScouts and MSAID. He has no operational role in either company. A.J. has received honoraria from AstraZeneca. C.H. has had an advisory role and received honoraria and research funding from Boehringer Ingelheim, Novartis, and Roche. S.F. has had an advisory role and received honoraria from Illumina. The remaining authors declare no competing interests.

## Additional information

Marcus Renner[1,2,3,50], Małgorzata Oleś [4,50], Nagarajan Paramasivam[4], Christoph E. Heilig [1,2,3], Annika Schneider[5,6], Caroline Modugno[7], Catherine Herremans[7], Jennifer Hüllein[4], Barbara Hutter [4], Cihan Erkut [3,8], Andreas Mock [6,9], Eva Krieghoff-Henning[1,2,10,11,12], Cecilia B. Jensen[5], Amirhossein Sakhteman[5], Matthew The [5], Tony Prinz [13,14,15,16], Panna Lajer[2,17], Annika Baude-Müller[1,2,3], Katja Beck[1,2,3], Bettina Beuthien-Baumann[18], Leonidas Apostolidis [2,19], Sebastian Bauer [20,21], Melanie Boerries [22,23,24], Christian H. Brandts[25,26,27,28], Damian T. Rieke [29,30], Thomas Kindler[31,32,33], Frederick Klauschen [6,9,34], Klaus Schulze-Osthoff[35,36], Richard F. Schlenk [2,3,19,37,38], Guy Berchem[7], Michael Allgäuer [39], Gunhild Mechtersheimer[39], Albrecht Stenzinger [39], Daniel B. Lipka [2,3,17], Matthias Schlesner [40], Bernhard Kuster [5,6], Arne Jahn [13,14,15,16,41], Evelin Schröck[13,14,15,16,41,42], Christoph Heining[15,43,44], Maria-Veronica Teleanu[1,2,3], Peter Horak [1,2,3], Simon Kreutzfeldt[1,2,3], Daniel Hübschmann [3,4,45,46], Wolfgang Hartmann [47], Hanno Glimm [15,43,44,48,51] & Stefan Fröhling [1,2,3,49,51] ✉

[1]Division of Translational Medical Oncology, German Cancer Research Center (DKFZ), Heidelberg, Germany. [2]National Center for Tumor Diseases (NCT), NCT Heidelberg, a partnership between DKFZ and Heidelberg University Hospital, Heidelberg, Germany. [3]German Cancer Consortium (DKTK), Core Center Heidelberg, Heidelberg, Germany. [4]Computational Oncology Group, Molecular Precision Oncology Program, NCT Heidelberg, Heidelberg, Germany. [5]School of Life Sciences, Technical University Munich, Freising, Germany. [6]DKTK, Partner Site Munich, Munich, Germany. [7]Service d'Hémato-Oncologie, Centre Hospitalier de Luxembourg, Luxembourg City, Luxembourg. [8]Division of Applied Functional Genomics, DKFZ, Heidelberg, Germany. [9]Institute of Pathology, Ludwig Maximilians University Munich, Munich, Germany. [10]Division of Personalized Medical Oncology, DKFZ, Heidelberg, Germany. [11]Department of Personalized Oncology, DKFZ-Hector Cancer Institute, University Medical Center Mannheim, Medical Faculty Mannheim, Heidelberg University, Mannheim, Germany. [12]Department of Personalized Oncology, University Medical Center Mannheim, Medical Faculty Mannheim, Heidelberg University, Mannheim, Germany. [13]Institute for Clinical Genetics, Faculty of Medicine and University Hospital Carl Gustav Carus, Dresden University of Technology (TUD), Dresden, Germany. [14]European Reference Network on Genetic Tumour Risk Syndromes, Hereditary Cancer Syndrome Center Dresden, Dresden, Germany. [15]DKTK, Partner Site Dresden, Dresden, Germany. [16]DKFZ, Heidelberg, Germany. [17]Section of Translational Cancer Epigenomics, Division of Translational Medical Oncology, DKFZ, Heidelberg, Germany. [18]Division of Radiology, DKFZ, Heidelberg, Germany. [19]Department of Medical Oncology, Heidelberg University Hospital, Heidelberg, Germany. [20]Department of Medical Oncology, West German Cancer Center, University Hospital Essen, Essen, Germany. [21]DKTK, Partner Site Essen, Essen, Germany. [22]Comprehensive Cancer Center Freiburg, University of Freiburg Medical Center, Faculty of Medicine, University of Freiburg, Freiburg, Germany. [23]Institute of Medical Bioinformatics and Systems Medicine, University of Freiburg Medical Center, Faculty of Medicine, University of Freiburg, Freiburg, Germany. [24]DKTK, Partner Site Freiburg, Freiburg, Germany. [25]University Cancer Center (UCT) Frankfurt, University Hospital Frankfurt, Goethe University, Frankfurt, Germany. [26]Department of Medicine, Hematology/Oncology, University Hospital Frankfurt, Goethe University, Frankfurt, Germany. [27]Frankfurt Cancer Institute, Frankfurt, Germany. [28]DKTK, Partner Site Frankfurt, Frankfurt, Germany. [29]Charité Comprehensive Cancer Center, Charité – Universitätsmedizin Berlin, Berlin, Germany. [30]DKTK, Partner Site Berlin, Berlin, Germany. [31]UCT Mainz, Johannes Gutenberg University Mainz, Mainz, Germany. [32]Department of Hematology and Medical Oncology, University Medical Center, Mainz, Germany. [33]DKTK, Partner Site Mainz, Mainz, Germany. [34]Berlin Institute for the Foundations of Learning and Data, Berlin, Germany. [35]Department of Molecular Medicine, Interfaculty Institute for Biochemistry, University of Tübingen, Tübingen, Germany. [36]DKTK, Partner Site Tübingen, Tübingen, Germany. [37]Department of Hematology, Oncology and Rheumatology, Heidelberg University Hospital, Heidelberg, Germany. [38]Clinical Trial Center, NCT Heidelberg, Heidelberg, Germany. [39]Institute of Pathology, Heidelberg University Hospital, Heidelberg, Germany. [40]Biomedical Informatics, Data Mining and Data Analytics, Faculty of Applied Computer Science and Medical Faculty, University of Augsburg, Augsburg, Germany. [41]NCT, NCT/University Cancer Center (UCC) Dresden, a partnership between DKFZ, Faculty of Medicine and University Hospital Carl Gustav Carus, TUD, and Helmholtz-Zentrum Dresden-Rossendorf, Dresden, Germany. [42]Max Planck Institute of Molecular Cell Biology and Genetics, Dresden, Germany. [43]Department of Translational Medical Oncology, NCT, NCT/UCC Dresden, a partnership between DKFZ, Faculty of Medicine and University Hospital Carl Gustav Carus, TUD, and Helmholtz-Zentrum Dresden-Rossendorf, Dresden, Germany. [44]Translational Medical Oncology, Faculty of Medicine and University Hospital Carl Gustav Carus, TUD, Dresden, Germany. [45]Innovation and Service Unit for Bioinformatics and Precision Medicine, DKFZ, Heidelberg, Germany. [46]Pattern Recognition and Digital Medicine Group, Heidelberg Institute for Stem Cell Technology and Experimental Medicine, Heidelberg, Germany. [47]Gerhard Domagk Institute of Pathology, Münster University Hospital, Münster, Germany. [48]Translational Functional Cancer Genomics, DKFZ, Heidelberg, Germany. [49]Division of Translational Precision Medicine, Institute of Human Genetics, Heidelberg University, Heidelberg, Germany. [50]These authors contributed equally: Marcus Renner, Małgorzata Oleś. [51]These authors jointly supervised this work. Hanno Glimm, Stefan Fröhling. ✉e-mail: stefan.froehling@nct-heidelberg.de

