## [Transparent Peer Review file · Nature Communications]

Multi-Layered Molecular Profiling Informs the Diagnosis and Targeted Therapy of Desmoplastic Small Round Cell Tumor

Corresponding Author: Professor Stefan Fröhling

Version 1:

Reviewer comments:

Reviewer #1

(Remarks to the Author)

This paper studied the multi omics data of DSRCT. Although the number of cases is not large, considering that it is a rare tumor, it still has certain scientific significance. There are the following modification suggestions.

1. Please state the statistical power for 30 patients with advanced DSRCT, addressing potential selection bias (e.g., heterogeneity in prior treatments may influence outcomes).
2. Please expand on why ERBB2 was prioritized over other targets ?
3. Please define "ERBB2 overexpression" criteria (e.g., fold-change and Benjamini-Hochberg adjusted P-value). In particular, in Figure 4d, the FC of ERBB2 protein was 0.67, which cannot be defined as a high level of expression.
4. In the differential expression analysis, the authors detected high levels of ERBB2 and CLDN6 in all and two of nine DSRCT patients, respectively. Is this result reproducible? especially given the small sample size (n=9).
5. The subheading titles in the Results section (e.g., 'DNA analysis,' 'RNA analysis,' 'Proteome and phosphoproteome analysis') are too generic and methodological. They should be revised to summarize key findings or biological insights, helping readers grasp the significance of each section.

Reviewer #2

(Remarks to the Author)

The manuscript by Renner and Oles et al. entitled 'Multi-layered molecular profiling informs the diagnosis and targeted therapy of desmoplastic small round cell tumor' (DSRCT) reports on a relatively small cohort of patients with refractory DSRCT that underwent molecular profiling to identify potentially actionable targets within a German molecular precision oncology program. The authors report that 13 of 30 patients received the recommended therapy that led to temporary disease control. The patients were treated with pazopanib and/or trastuzumab deruxan, and the authors claim that the observed clinical responses were 'triggered by ERBB2 overexpression in the absence of constitutive ERBB2 signaling'. The main conclusion of this paper is that 'multi-omics profiling enables individualized DSRCT treatment', which is expected and a rather generic conclusion.

While the paper is well-written and easy to follow, the main concerns stem from the facts that the paper is rather descriptive in nature, rather appearing like a clinical case series, and that the main conclusions are obvious. The fact that ERBB2 overexpression can be used as a predictive marker and target for therapy for cancer in general is well-established across cancer entities and not particularly novel for DSRCT. Several earlier reports have demonstrated this for DSRCT: 'HER2 Antibody-Drug Conjugates Are Active against Desmoplastic Small Round Cell Tumor', Zhang et al. 2024 Clin Can Res; 'High expression level of ERBB2 and efficacy of trastuzumab deruxtecan in desmoplastic small round cell tumour: a monocentric case series report', Brahmi et al. 2025 ESMO Open; 'Off-label use of fam-trastuzumab deruxtecan-nxki with early activity in a cohort of patients with desmoplastic small round cell tumor', Slotkin et al. 2024 ASCO Annual Meeting. These studies were based on even earlier work from Wu et al. (Multi-site desmoplastic small round cell tumors are genetically related and immune-cold, 2022 NPJ Precision Oncology), who proposed ERBB2 as a suitable target for a subset

of DSRCT patients. Likewise, the possibility of treating DSRCT patients with pazopanib is not novel: 'Pazopanib in advanced desmoplastic small round cell tumours: a multi-institutional experience', Frezza et al. 2014 Clinical Sarcoma Research; 'Clinical Activity of Pazopanib in Patients with Advanced Desmoplastic Small Round Cell Tumor' Ludwig et al. 2018 Oncologist.

To this reviewer, the most interesting part of this paper is the availability of the multiomics data. Unfortunately, the link and accession code provided to not give access to the data (yet). Another valuable point of this paper is that it is intriguing that 8 of 30 patients were misdiagnosed initially, which underscores that DSRCT is a diagnostic challenge and that new diagnostic biomarkers are required. In this context, the authors should include in the discussion section apart their appraisal of multiomics also other new diagnostic markers, such as CACNA2D2. Further, the authors should investigate the functional mechanism underlying the overexpression of ERBB2 and other proposed targets, such as SSTR3/5 and CLDN6, as well as their functional role(s) for the malignant phenotype of DSRCT. Otherwise, this descriptive report does not meet the bar for publication in such a prestigious journal, and appears to be more suitable for a specialized/clinical journal.

Reviewer #3

(Remarks to the Author)

The authors report new findings from the MASTER programme conducted by the German Cancer Research Center (DKFZ) in Heidelberg, focused on using multi-omics profiling to enable personalised treatment recommendations for rare, aggressive cancers (DSRCT). This study expanded on previous omics-based approaches and combined genome/exome sequencing, RNA sequencing and DNA methylation profiling and proteomics and phosphoproteomics. This approach is a first in its breadth of components it combined, and found promising signals that a multi-layered molecular profiling can both correct wrong or incomplete diagnoses and provide effective personalised to slow or halt disease progression. These findings are of great interest to the oncological as well as the wider scientific community, as currently no effective standardised treatment regime for DSRCT exists, and a significant percentage of patients are misdiagnosed or receive a delayed diagnosis. As Nature Communications aims at a broader readership in contrast to purely clinical journals, revising some of the language and detailing both the methodology and the implications of the findings may improve accessibility.

This was an exploratory study, which was not powered appropriately to test for treatment superiority, therefore the authors make no claims of statistical significance. The overall study conduct was appropriate for the research objectives, however I have noted several questions regarding study specifics, timeline and recruitment criteria.

No statistical analysis was conducted, as this was not a powered superiority study. The authors report findings suitably with summary statistics, proportions and confidence intervals were appropriate. The data created for this study has been reported in full and is also available online. Full personalised treatment recommendations, DNA, RNA analyses etc. are available as supplementary information. There have been a few small clerical errors I referenced in the commented manuscript.

I share the authors' conclusion, that the study provides evidence for both diagnostic and therapeutic improvements over the current standard of care. The authors have backed up their conclusion with specific numbers on how much of the treatment recommendations was attributable to the additional tests and screening. There is considerable reference to other studies, which is important to compare and give background to the study's findings. However, the detail given is somewhat diluting the novel results of the study. Perhaps some of the detail reported on the other studies can be shortened to keep the discussion section more focussed.

Relevant references have been used effectively throughout the manuscript to support statements and give background. In a few instances I have left comments asking for additional references.

As a biostatistician I am not qualified to assess the suitability of methods used for the sample processing and DNA, RNA and genome sequencing, and have therefore not commented on these aspects.

No pre-registration of the study protocol was reported, therefore no check of protocol deviations was performed. Ethics have been obtained prior to study beginning from the appropriate authorities, and was reported in the manuscript.

Suggestions on minor improvements are attached in the commented version of the manuscript. I would recommend accepting this manuscript subject to minor revisions.

Reviewer #4

(Remarks to the Author)

The manuscript by Renner et al. describes a multi-omics precision oncology approach for 30 patients with advanced desmoplastic small round cell tumor (DSRCT) enrolled in the German MASTER trial. Each patient's tumor underwent whole-genome/exome sequencing, RNA sequencing, DNA methylation, and proteomic/phosphoproteomic profiling, with results discussed by a molecular tumor board (MTB) to recommend personalized therapies. In 93% of patients (28/30), the MTB was able to issue at least one therapeutic recommendation (median 3–4 per patient). In total, 107 individual recommendations were made across various categories including tyrosine kinase inhibitors (45% of recommendations),

DNA damage repair (12%), immunotherapy (11%), theranostic approaches (10%), and others. Notably, 73% of all recommendations were based on RNA expression findings – for example, overexpression of certain tyrosine kinase (TK) pathway genes or cell-surface targets. The authors highlight that SSTR3/SSTR5 (somatostatin receptors) were over-expressed in about one-third of cases and CLDN6 in ~20% of cases, leading the MTB to frequently suggest peptide receptor radionuclide therapy (PRRT) or CLDN6-directed CAR T-cell therapy, respectively. Another key finding is that some DSRCT tumors showed elevated ERBB2 (HER2) mRNA expression without gene amplification, which prompted the MTB to recommend the HER2-targeted antibody–drug conjugate trastuzumab deruxtecan (T-DXd). Two patients who received off-label T-DXd experienced durable partial remissions (>12–15 months disease control), an impressive clinical outcome in this refractory sarcoma. The median overall survival from the first MTB consultation was reported as ~2.1 years, with a 4-year survival rate of ~10%, reflecting the aggressive nature of DSRCT despite heavy pretreatment. Overall, the study demonstrates that multi-layered molecular profiling can uncover actionable alterations in nearly every DSRCT patient, and that implementing MTB-recommended targeted therapies (such as multitarget TK inhibitors and novel ADCs) can lead to meaningful clinical responses in this ultra-rare, hard-to-treat sarcoma.

Major Comments

1. Clarify MTB Decision-Making Rationale: The manuscript would benefit from a clearer explanation of how the Molecular Tumor Board formulated its recommendations in each case. Currently, the results enumerate the number and types of recommendations, but the rationale behind each recommendation is not sufficiently described. For example, it is unclear if the MTB followed predefined actionability criteria (e.g. ESMO Scale of Clinical Actionability of molecular Targets) or a structured algorithm. The authors should expand on their decision-making process: what specific molecular findings (alone or in combination) were deemed actionable, and why? Were recommendations based on levels of evidence from literature or on internal consensus/expert opinion? Detailing a few representative case vignettes could be helpful – for instance, why a particular gene's overexpression or mutation led to a therapy recommendation, and how the MTB weighed other factors (prior therapies, patient condition, etc.) in that decision. Providing this information will make the process more transparent and reproducible. As an example, the text notes that 57% of patients were referred to clinical trials and that certain recommendations (e.g. PARP inhibition, immunotherapy) were sometimes made based on mutational signatures or expression alone; the manuscript should explicitly justify these choices. In summary, how did the MTB translate multi-omic data into a treatment plan? A brief methodological description (possibly referencing Supplementary Fig. 2a) of the MTB workflow and decision criteria would greatly strengthen the paper's clinical relevance.
2. Include Survival Analyses for MTB Impact: I suggest an important analysis currently missing: a comparison of clinical outcomes between patients who received MTB recommendations versus those who did not, and between patients who followed an MTB-recommended therapy versus those who did not. Such an analysis would address the utility of the MTB's guidance. For instance, only 28 of 30 patients received recommendations (two patients had none) – how did their outcomes compare? More critically, among the 28 with recommendations, only 13 patients actually received an MTB-recommended molecularly guided therapy (as per Table 2). Do these 13 patients exhibit any trend toward improved survival (or longer progression-free survival) compared to those who, for whatever reason, did not receive a recommended therapy? Even acknowledging selection biases (patients able to receive these therapies might have better performance status, etc.), presenting a Kaplan–Meier analysis or summary statistics for overall survival in these subgroups would be very informative. For example, one could compare median OS from MTB enrollment for “MTB-treated” patients vs. “non-MTB-treated” patients. If no significant difference exists, that is also worth discussing candidly. This comparative survival analysis would directly address the clinical impact of the MTB program – a key question for a reader. The lack of such analysis is noted by the reviewer, and adding it (even if descriptive due to small numbers) would strengthen the manuscript's claims that the MTB's recommendations confer meaningful benefit. If numbers are too small for formal statistics, the authors can still qualitatively comment on any survival differences. (For instance, it appears that patients who accessed recommended targeted therapies achieved some prolonged disease control as noted in specific cases, but a cohort-level comparison is needed to generalize this.)
3. Define Expression Thresholds for “Overexpression”: A major point of confusion is how the study defined a gene as “overexpressed” or having “increased mRNA expression.” Many of the MTB's recommendations – 78 of 107 (73%) – were based on elevated transcript levels of potential targets. However, the manuscript does not specify what thresholds or reference comparisons were used to call something “overexpressed.” The authors should clearly state the criteria for deeming a target's mRNA upregulated. For example, did they compare the tumor's transcripts per million (TPM) values against a reference dataset of other sarcomas or normal tissues? Was a specific fold-change or percentile rank used as a cutoff? The absence of such definitions leaves the reader wondering how targets like ERBB2, CLDN6, SSTR3, or SLFN11 were selected for action. The Supplementary Figures hint that the authors compared expression in DSRCT relative to other sarcoma types (e.g. Supplementary Fig. 4 shows SSTR family expression in DSRCT vs other sarcomas) – indeed, SSTR3 and SSTR5 were consistently higher in DSRCT than in other sarcomas. If this comparative approach was used generally, it should be stated. Similarly, the manuscript notes that the MTB recommended HER2-targeted therapy (T-DXd) in seven patients due to increased ERBB2 mRNA, and recommended CLDN6-targeted CAR-T in some patients due to CLDN6 expression. The exact cutoff (if any) for “high” ERBB2 or CLDN6 expression should be explained. Was any detectable CLDN6 in RNA-seq considered actionable (given CLDN6 is normally absent in adult tissues), or did it have to exceed a TPM threshold? For HER2, did the authors have an expression level equivalent to “HER2-low” or “HER2-high” as per known breast cancer criteria, or was it purely relative within their cohort? Likewise, SLFN11 expression: the text references “high SLFN11” in one patient prompting a PARP inhibitor recommendation, but how was “high” defined? Clarity on these points is crucial, because the concept of actionability based on RNA is relatively new – readers need to know how robustly overexpression was determined. The authors should also discuss the rationale for considering mere mRNA overexpression as sufficient to target, especially for cases like ERBB2 where no gene amplification or mutation was present (i.e. why presume that high transcript alone makes the protein a good target?). Overall, please add a Methods description or supplementary table detailing the expression-based target calling strategy (thresholds, comparisons, and any validation by

protein IHC if done). This will greatly enhance the transparency and reproducibility of the study's approach. 4. Provide Literature Context for SLFN11 and PARP Inhibitor: The recommendation of PARP inhibitor therapy in SLFN11-positive patients is biologically plausible, but the manuscript needs to better justify this with prior research. The authors note that one patient had increased SLFN11 expression (along with mutational signature SBS8) which led the MTB to advise a PARP inhibitor (olaparib) combined with trabectedin. They also mention up to 10 patients had recommendations for PARP inhibitors due to HR pathway deficiencies (SBS3/8 or BRCA-pathway mutations) and in some cases high SLFN11. However, the rationale for using SLFN11 as a biomarker of PARP inhibitor sensitivity should be explicitly stated. Prior studies (e.g. in Ewing sarcoma, small cell lung cancer, etc.) have shown that SLFN11 expression correlates with responsiveness to DNA-damaging agents and PARP inhibitors. If available, the authors should cite relevant literature to support this approach. For instance, it is known that SLFN11 interferes with replication fork recovery, making tumors with high SLFN11 more vulnerable to agents like platinum or PARP blockade. By including a sentence in the Discussion such as, "This recommendation was motivated by reports that SLFN11-high tumors are especially sensitive to PARP inhibition," the authors can ground their MTB decision in published evidence. Additionally, if the patient who received olaparib (DSRCT-14 in Table 2) achieved stable disease for 4 months, that outcome could be contextualized by noting it aligns with the hypothesized role of SLFN11. In summary, please add a brief discussion of why SLFN11-positive status justified PARP inhibitor therapy, with references to prior studies of SLFN11 as a predictive biomarker. This will help readers understand the biological reasoning and novelty of applying that rationale in DSRCT.

5. Clarify Basis for Recommending Immunotherapy: The manuscript mentions that 12 of 107 recommendations (11%) fell into the "IT" (immunotherapy) basket, including suggestions such as PD-1 checkpoint blockade and CLDN6-directed CAR T-cells (the latter could also be considered cell therapy). However, it is unclear what criteria were used to recommend immune checkpoint inhibitors in this cohort. DSRCT is generally not known as an immunologically "hot" tumor, so on what basis did the MTB suggest PD-1 inhibitor therapy (e.g., nivolumab for patient DSRCT-03, per Table 2)? Did any patients have biomarkers like high tumor mutation burden, mismatch-repair deficiency, PD-L1 expression, or prominent T-cell infiltrates? The supplementary data (Table 6) intriguingly shows one patient (DSRCT-19) with mutational signature SBS2 who was given an "immune checkpoint" recommendation. SBS2 is an APOBEC-related signature; the connection between SBS2 and immunotherapy efficacy is not obvious. If the authors used any immune gene expression signatures (for example, a tertiary lymphoid structure [TLS] signature or cytotoxic T-cell signature from the RNA-seq) or any specific immunologic classifier to identify patients who might benefit from immunotherapy, this should be stated. The reviewer specifically asks whether TLS gene signatures were used – since TLS presence has been correlated with immunotherapy response in some sarcomas, did the authors evaluate this or other metrics (like the "immune hot/cold" expression clusters)? If not, the authors should clarify that immunotherapy was recommended on more general grounds (e.g., last-line option or extrapolation from other sarcomas) rather than on a clear predictive biomarker. In the revised manuscript, please explain the rationale for immunotherapy recommendations: was it based on any molecular/immunological evidence or simply a trial enrollment opportunity? For transparency, list what factors (if any) led to recommending PD-1/PD-L1 blockade in those cases. This will preempt readers' questions about whether those suggestions were driven by data (for example, "Patient X had moderate CD8+ T cell infiltration and PD1 gene expression, justifying nivolumab trial") or whether they were more empirical. Such clarification is important, given the current lack of approved immunotherapy indications in DSRCT.

6. Contextualize Novelty and Prior Literature on Targeted Therapies in DSRCT: The study's findings should be better positioned within the context of existing literature, to highlight both consistency with known results and the novel contributions. The authors are investigating an ultra-rare sarcoma where little targeted therapy data exists, so any connections to prior studies should be emphasized. For instance, pazopanib (multi-kinase inhibitor) has been used off-label in DSRCT before – a retrospective series of 29 patients (Ludwig et al., 2018) showed disease stabilization in ~55% and a median PFS of ~5.6 months. In the manuscript's Table 2, multiple patients received pazopanib due to various TK gene overexpressions, with several achieving partial responses or stable disease lasting 6–17 months, which is quite encouraging and generally in line with the prior report. Citing such studies would show that the authors are building on a foundation (e.g., using pazopanib as a baseline known active drug in DSRCT) and then going further by personalizing its use to those with specific angiogenic/kinase signals. The most striking novel finding here is the efficacy of HER2-targeted ADC (T-DXd) in DSRCT. This is, to my knowledge, the first formal report of clinical responses to an ADC in DSRCT patients, supported by the authors' multi-omics rationale. The Introduction references preclinical evidence that Her-family signaling could be a vulnerability in DSRCT, including a report that afatinib, cetuximab, and T-DXd reduced DSRCT xenograft growth. The authors should explicitly state that their two patient cases on T-DXd represent a translational validation of those preclinical findings. In fact, a very recent case series (likely emerging in 2024–2025) has also suggested that HER2 is a therapeutic target in DSRCT and that T-DXd can benefit these patients – the authors may want to reference this if available, as it underscores the relevance of their approach. Overall, the Discussion should be refined to stress the study's novelty: multi-layered molecular profiling in DSRCT has not been done at this scale before, and the results reveal previously unappreciated therapeutic options (e.g., PRRT for SSTR3-positive disease, CLDN6 CAR-T for CLDN6-expressing tumors, ADC for HER2-low tumors). At the same time, the findings are consistent with emerging paradigms in rare cancers, where RNA or proteomic data can identify actionable targets even in the absence of classic genomic alterations. Emphasizing this consistency and novelty will help convince readers (and the editors) that the manuscript is a significant advance worthy of publication.

Minor Comments

Gene Expression Cutoffs in Text: When describing expression-based findings, the manuscript currently uses qualitative terms like "overexpression" or "increased expression" without numeric detail. In revision, consider reporting the actual expression levels for key targets in a comparative sense. For example: "ERBB2 was expressed at 70 TPM in Patient 28's tumor, which is >4× the median level in our DSRCT cohort. Similarly, when stating "consistent overexpression of SSTR3 and SSTR5 in DSRCT compared to other sarcomas", it would help to quantify ("median TPM X vs Y in other sarcomas") either in text or a supplemental table. This would complement the needed clarification of thresholds (as per Major Comment #3).

Terminology – “Constitutive signaling”: The phrase “ERBB2 overexpression in the absence of constitutive ERBB2 signaling” might be confusing to readers. It appears to mean that no activating mutation or high phosphorylation of ERBB2 was present (i.e., no evidence of strong signaling activity), yet the gene was highly transcribed. The authors might rephrase this for clarity. Perhaps specify that no ERBB2 kinase activation was detected via phosphoproteomics, supporting the use of an ADC (which requires surface expression but not signaling). This distinction between targeting a protein for its expression vs. targeting an active driver should be made clear to avoid misinterpretation.

Proteomics Results – Integration: The paper mentions proteomic/phosphoproteomic profiling was performed in 9 patients, and gives a few examples (e.g., FGFR4, KDR phosphorylation in certain cases). However, it’s not entirely clear how these proteomic findings were used. The authors should ensure they close the loop on those results. For instance, patient DSRCT-18 had activated KDR and TYRO3 by phosphoproteomics and did respond to pazopanib (which hits VEGFR/KDR) for 17 months – this is a compelling connection that could be highlighted as a success of multi-omics integration.

Conversely, the lack of ERBB2 phosphorylation in the two T-DXd-treated patients is noted as justification for choosing an ADC over a small-molecule inhibitor. These points do appear in the text, but the authors might make them more explicit so that the contribution of proteomics is appreciated. If proteomic data did not significantly change recommendations (beyond confirming known targets), that could be stated too, as it teaches where multi-omics adds value.

Formatting and Typographical Issues: Overall the manuscript is well-written. We caught a few minor formatting issues that should be corrected:

In Supplementary Table 3 (PDF), the percentage “12,1%” is written with a comma; this should be “12.1%” in English decimal format. Please ensure consistency in numeric formatting (use a period for decimals in the main text and supplements).

There are instances of awkward line breaks in the PDF (e.g., the radioisotope fluorine-18 in 18F-FDG-PET appears split across lines in our copy). In the final production, such issues will likely be fixed by copyeditors, but the authors should double-check the rendering of all technical terms and units (especially in figure labels).

All acronyms should be defined at first use. For example, “MTT” is used in Figure 2 legend (for “molecularly targeted treatment”) – ensure this is defined in the text or figure caption so readers know it refers to treatments given outside trials based on MTB recommendations

Gene symbols are appropriately capitalized; just verify if the journal prefers them italicized. In Nature journals, human gene symbols are typically written in roman (non-italic) uppercase, which seems to be what the authors have done (e.g., WT1, EWSR1 in text). This is fine, but ensure consistency (e.g., in one case “EWSR1::WT1” fusion was clearly described; make sure all gene fusion notations use the same format throughout).

Minor grammar: page 2 of the PDF has author affiliations where a name is split by a line break (“Annika BaudeMüller”) – this is just a PDF artifact, not an issue for content. No significant typos were noted in the main text.

Data Availability and Figures: The authors should double-check that all figures and tables are properly referenced in the text in order. For instance, Figure 5e is cited when discussing ERBB2 expression and IHC – make sure Figure 5e indeed shows what is described (ERBB2 mRNA and protein levels). Similarly, ensure that each supplementary figure/table is cited at the appropriate point. A quick check suggests everything is cited, but it’s worth verifying consistency one more time (e.g., Supplementary Figure 2a is mentioned as containing the multi-omic biomarker overview so Supplementary Fig. 2 should indeed illustrate that workflow). Also, minor point: in Table 2, the footnotes A, B, C are very useful to explain combination treatments – ensure these footnotes are included in the main text or legend for clarity (they appear to be in the PDF bottom of Table 2).

By addressing the above comments – especially the major points about methodology clarity and additional analyses – the authors can substantially improve the manuscript’s clarity and impact.

Version 2:

Reviewer comments:

Reviewer #1

(Remarks to the Author)

The authors have adequately addressed all major concerns raised in my initial review. The additional data from 11 DSRCT samples further enhance the reproducibility of the proteomic findings. The authors should note the overlapping labels for DSRCT and MPNST in the legend of Figure 4a.

Reviewer #2

(Remarks to the Author)

Thank you for the careful revisions and for sharing additional context. I agree the manuscript is clearly written and the real-world, registry-style approach is valuable in an ultra-rare cancer. That said, several core claims on novelty and mechanistic insight still feel overstated relative to what is already known, both in DSRCT specifically and, more broadly, in precision oncology.

First, the assertion that your series provides the first patient-level evidence that ERBB2 can be therapeutically exploited in DSRCT is no longer tenable. Prior work had already proposed and supported ERBB2 as a target in DSRCT at the transcript/protein level and in patient-derived models (Zhang et al. 2024 Clin Cancer Res), and, crucially, independent clinical reports contemporaneous with your submission describe responses to trastuzumab deruxtecan (T-DXd) in DSRCT (Brahmi et al. 2025 ESMO Open; Slotkin et al. 2025 ASCO). These publications collectively show high ERBB2 expression in DSRCT and early clinical activity of T-DXd, substantially narrowing the space for claims of therapeutic novelty.

Relatedly, the manuscript’s framing that responses were “triggered by ERBB2 overexpression in the absence of constitutive

ERBB2 signaling” risks implying a new principle for ADC treatment selection. By design, HER2-directed ADCs such as T-DXd do not require constitutive kinase signaling; they require sufficient cell-surface target for binding, internalization, and payload delivery. This signaling-agnostic mechanism is well established across tumor types and underpins the clinical success of T-DXd in HER2-low disease (Modi et al. 2022 NEJM) and the FDA’s tumor-agnostic accelerated approval for HER2-expressing solid tumors in 2024. Thus, the absence of phospho-ERBB2 is not a mechanistic justification uniquely supporting ADC use in DSRCT—if anything, it simply aligns DSRCT with the broader, already-codified pharmacology of ADCs. I encourage you to temper the mechanistic narrative accordingly.

On ERBB2 biology per se, your phosphoproteomic snapshots are interesting, but I remain cautious about elevating “lack of constitutive activation” to a definitive pathway conclusion. Even recent methodologic benchmarks emphasize that kinase-activity inference from phosphoproteomics carries nontrivial uncertainty and context dependence (Müller-Dott et al. 2025 Nat Commun). These data are hypothesis-generating—not dispositive—and they do not materially alter the existing rationale that T-DXd efficacy is largely independent of ERBB2 signaling status.

Second, the therapeutic observations with pazopanib are clinically useful but not new in DSRCT. Multi-institutional and single-center series—spanning a decade—have already documented responses and disease control with pazopanib in pretreated DSRCT (Frezza et al. 2014 Clin Sarcoma Res; Menegaz et al. 2018 The Oncologist). Your cohort adds confirmatory, real-world experience but does not establish predictive biomarkers for pazopanib beyond exploratory correlations; here again, the claim of innovation should be moderated.

Third, several “novel” targets highlighted—SSTR family members and CLDN6—have been discussed previously as plausible DSRCT or pediatric solid-tumor targets. SSTR-based imaging/theranostics are long-standing in neuroendocrine oncology and have sporadically been explored across sarcomas; their mention in DSRCT is therefore incremental rather than de novo. CLDN6 is an active pediatric-solid-tumor target with multiple first-in-human ADC/CAR-T programs, and recent cross-entity datasets document CLDN6 expression in subsets of pediatric tumors, including DSRCT—again arguing for caution in portraying CLDN6 as a newly uncovered opportunity here (Tsang et al. 2024 AACR poster; Seidmann et al. March 2025 Cancers).

Fourth, on scope and impact: a 30-patient prospective series is certainly commendable in an ultra-rare disease, but it is best framed as a well-executed clinical case series within an established precision-oncology paradigm rather than a first demonstration that “multi-omics enables individualized treatment.” Large prospective programs, among them the same from which the current DSRCT data came from, have already shown that comprehensive genomic/transcriptomic profiling can guide therapy and produce meaningful clinical benefit in rare cancers broadly (MASTER / DKTK-NCT: Horak et al. 2021 Cancer Discovery), with multi-omics and structured MTB workflows now commonplace in national networks. Your work is important because it applies this model to DSRCT with deeper proteomic layers—not because it newly proves the overarching concept. Clarifying this positioning would strengthen the manuscript.

Fifth, I appreciate the emphasis on misdiagnosis. This is indeed a practical pain point in DSRCT, and the field now has emerging, disease-specific diagnostics. In particular, the recent CACNA2D2 study provides strong evidence for a single, robust immunohistochemical marker to distinguish DSRCT from its morphological mimics (Geyer et al. 2025 Cancer Communications). The authors refer to this work in the revised manuscript, but I recommend briefly discussing it work alongside methylation-based sarcoma classifiers (Koelsche et al. 2021 Nat Commun; Jäger et al. 2025 bioRxiv) to situate your diagnostic narrative within current tools. One could ask the question why complex, time-consuming and highly specialized multi-omics profiling is necessary to faithfully diagnose DSRCT if a simple IHC marker with/without auxiliary DNA methylation profiling can lead to the same outcome?

Finally, a note on how conclusions are phrased. The current text tends to conflate three different contributions: (i) confirmatory evidence that ERBB2 is frequently expressed in DSRCT; (ii) registry-style documentation that T-DXd and pazopanib can achieve disease control in selected patients; and (iii) exploratory phosphoproteomic and proteomic readouts. Each is valuable, but only (ii) is truly patient-level clinical evidence; (i) is now well supported by prior literature; and (iii) remains hypothesis-generating. I would encourage a more restrained take-home message, for example: “In a prospective real-world registry, multi-layer profiling refined diagnosis in DSRCT and supported rational use of established targeted agents (including T-DXd and pazopanib), warranting biomarker-enriched prospective trials.” This would keep the paper’s strengths front and center—cleanly curated clinical data, thoughtful integration, and transparent follow-up—without overreaching on novelty.

Concrete suggestions to align claims with evidence:

Recast ERBB2/T-DXd content as confirmatory and contextualize with existing preclinical and clinical reports in DSRCT (Zhang et al. 2024; Brahmi et al. 2025; Slotkin et al. 2025). Emphasize your longer follow-up rather than “first-in-patients.”

Soften mechanistic language that infers pathway quiescence from phosphoproteomics; acknowledge methodological limits (Müller-Dott et al. 2025 Nat Commun).

Frame pazopanib results as real-world replication of prior activity and present biomarker analyses as exploratory (Frezza et al. 2014; Menegaz et al. 2018).

Update the diagnostic section to include CACNA2D2 and the role of methylation classifiers in reducing misclassification (Geyer et al. 2025; Koelsche et al. 2021, Jäger et al. 2025).

Position the study as an application/extension of an established multi-omics precision-oncology framework rather than an inaugural demonstration of clinical utility (Horak et al. 2021).

In sum, this is a valuable and carefully assembled case series. Tightening the novelty claims—particularly around ERBB2/T-DXd and the interpretation of phosphoproteomics—and anchoring your conclusions to what is uniquely contributed here (prospective, real-world integration and thoughtfully documented outcomes in an ultra-rare sarcoma) will, in my view, make the paper both more accurate and more compelling.

Reviewer #3

(Remarks to the Author)

I am satisfied with the changes made to the manuscript in response to the annotations in the manuscript I left. I recommend acceptance of the submission for publication.

Reviewer #4

(Remarks to the Author)

The authors have addressed my comments. I recommend acceptance for publication

NCOMMS-25-13803A

Multi-Layered Molecular Profiling Informs the Diagnosis and Targeted Therapy of Desmoplastic Small Round Cell Tumor

We sincerely thank the reviewers and editors for their thorough and thoughtful evaluation of our manuscript. We greatly appreciate the time and effort devoted to providing constructive feedback, which has helped us significantly strengthen the quality and clarity of our work. In response, we have carefully revised the manuscript and addressed each of the reviewers' comments in detail. Below, we provide a point-by-point response outlining the changes made and the rationale behind them.

Reviewer #1

This paper studied the multi omics data of DSRCT. Although the number of cases is not large, considering that it is a rare tumor, it still has certain scientific significance. There are the following modification suggestions.

1. Please state the statistical power for 30 patients with advanced DSRCT, addressing potential selection bias (e.g., heterogeneity in prior treatments may influence outcomes).

Response: We appreciate the reviewer's attention to this important point. As noted by another reviewer with biostatistical expertise, this was a hypothesis-generating, exploratory study that was not designed or powered to test for treatment superiority. Given the ultra-rare nature of DSRCT (incidence, $\sim 0.2/1,000,000$ persons/year), a cohort of 30 patients represents a substantial and clinically meaningful dataset in this context. However, due to the absence of a standard treatment approach and the advanced disease stage of all patients at enrollment, treatment histories and subsequent therapies were heterogeneous, reflecting the real-world clinical dilemma faced by physicians managing this disease. In light of this complexity, formal statistical power calculations are not applicable. Instead, we focused on reporting outcomes using summary statistics, proportions, and confidence intervals where appropriate. To avoid any misunderstanding, we have revised the manuscript to use the term "significant" exclusively in reference to statistical significance and have otherwise reworded statements to reflect descriptive findings. We have also ensured that the revised manuscript clearly states the nature of the study.

2. Please expand on why ERBB2 was prioritized over other targets?

Response: We appreciate the reviewer's thoughtful question. We would like to clarify that ERBB2 was not prioritized a priori over other potential targets. As part of our multi-omics profiling approach, all potentially actionable molecular alterations across DNA, RNA, and proteomic layers are systematically evaluated by the molecular tumor board (MTB). Treatment recommendations are made based on a structured and standardized decision-making process, which includes assigning evidence levels according to a nationally established framework (see also our responses to Reviewers #3 and #4 and references therein: Leichsenring et al., 2019; Horak et al., 2021; Horak et al., 2022; Mock et al., 2023). The final choice of therapy is guided not only by the molecular findings but also by additional clinical and practical factors, including the patient's general

condition and comorbidities, accessibility of drugs (especially relevant for off-label therapies) and clinical trials, as well as patient preference.

In the specific case of ERBB2, comprehensive profiling consistently revealed expression at both the mRNA and protein levels across most DSRCT cases. Several factors support the therapeutic recommendation of trastuzumab deruxtecan (T-DXd): (i) emerging preclinical data indicating efficacy of T-DXd in DSRCT models, (ii) clinical evidence from other tumor types showing that T-DXd can be effective even in tumors with low ERBB2 expression, and (iii) our phosphoproteomic data demonstrating that ERBB2 is not constitutively activated in DSRCT. This last point favors T-DXd over agents whose mechanism of action depends on ERBB2 kinase activity or dimerization, such as neratinib, afatinib, or pertuzumab, which have been previously suggested as potential therapies for DSRCT. Furthermore, the marked and durable response observed in our first patient treated with T-DXd provided additional clinical rationale to offer this therapy to other ERBB2-expressing patients, including those with lower expression levels.

In line with suggestions from this and other reviewers, we have expanded the revised manuscript to provide a more precise explanation of the MTB's decision-making process. We hope that this additional context clarifies the rationale for the therapeutic recommendations and the specific role of ERBB2-targeted therapy in this study.

3. Please define “ERBB2 overexpression” criteria (e.g., fold-change and Benjamini-Hochberg adjusted P-value). In particular, in Figure 4d, the FC of ERBB2 protein was 0.67, which cannot be defined as a high level of expression.

Response: We thank the reviewer for this helpful question and the opportunity to clarify our definition of “ERBB2 overexpression” at both the RNA and protein levels. As a general principle, expression – whether at the transcript or protein level – is always determined relative to a reference cohort.

For genes listed in the *Cancer Gene Census* (<https://www.sanger.ac.uk/data/cancer-gene-census>) as well as in our in-house lists of druggable genes and other genes of interest, RNA overexpression was defined as a fold change of more than two times the median of a comparison cohort of 148 RNA-sequenced samples within the MASTER program. Since October 2017, we have also applied an alternative criterion: for genes never found to be upregulated by a factor of 2, a z-score greater than 2 was considered indicative of overexpression.

The ERBB2 protein expression value cited by the reviewer (\log_{10} intensity difference of 0.67) reflects a high expression level according to our metric, with a mean \log_{10} intensity for DSRCT of 8.01 compared to 7.33 for a pan-sarcoma reference cohort. Recognizing that log-transformed fold changes are not always intuitive, we converted this value to the linear scale, yielding an approximate fold change of 4.97. In proteomic analyses, a fold change greater than 2 (or less than 0.5 for downregulation) is commonly used as a practical threshold for notable differential expression. The statistical significance of ERBB2 overexpression was assessed using Welch's t-test (accounting for unequal sample sizes), as reported in Figure 4d.

For the large-scale differential protein expression analysis shown in Figure 4c, we applied Benjamini-Hochberg correction to control the false discovery rate (FDR). This is standard practice

in large-scale proteomics to limit the proportion of false positives. The combination of FDR-adjusted p-values and fold-change information allows the identification of proteins with both statistical significance and meaningful effect size.

In response to the reviewer's comment, we have revised the Methods section and figure legends to make these definitions, statistical approaches, and the use of reference cohorts explicit and to present the protein expression differences in both log and linear scales for improved interpretability.

4. In the differential expression analysis, the authors detected high levels of ERBB2 and CLDN6 in all and two of nine DSRCT patients, respectively. Is this result reproducible? especially given the small sample size (n=9).

Response: We appreciate the reviewer's thoughtful comment. We fully agree that the reproducibility of proteomic and phosphoproteomic findings is crucial, particularly given the small size of the initial cohort and the clinical relevance of the results. Since the data cut-off used for the analyses presented in the manuscript, an additional 11 DSRCT patients have been enrolled in the MASTER program for whom (phospho)proteomic data are now available. Analysis of these additional data confirms the robustness of our initial findings (see Figure below): ERBB2 expression was consistently observed across all newly profiled cases, while CLDN6 expression was detected in approximately 25% of patients, closely matching the frequency in the original cohort. Moreover, in a prescreening of 10 sarcoma patients by immunohistochemistry, a single case, a DSRCT, showed high CLDN6 expression, which enabled the patient's enrollment in the BNT211-01 phase 1 trial (Mackensen et al., *Nat Med* 2023, PMID: 37872225). Together, these results underscore the reproducibility and broader relevance of our observations, further strengthening the clinical rationale for targeting these molecules in DSRCT.

Figure. ERBB2 and CLDN6 protein expression, as measured by mass spectrometry, in a validation cohort of 11 DSRCT patients. FC, fold change.

5. The subheading titles in the Results section (e.g., 'DNA analysis,' 'RNA analysis,' 'Proteome and phosphoproteome analysis') are too generic and methodological. They should be revised to

summarize key findings or biological insights, helping readers grasp the significance of each section.

Response: We agree and have amended the subheadings accordingly.

Reviewer #2

The manuscript by Renner and Oles et al. entitled ‘Multi-layered molecular profiling informs the diagnosis and targeted therapy of desmoplastic small round cell tumor’ (DSRCT) reports on a relatively small cohort of patients with refractory DSRCT that underwent molecular profiling to identify potentially actionable targets within a German molecular precision oncology program. The authors report that 13 of 30 patients received the recommended therapy that led to temporary disease control. The patients were treated with pazopanib and/or trastuzumab deruxan, and the authors claim that the observed clinical responses were ‘triggered by ERBB2 overexpression in the absence of constitutive ERBB2 signaling’. The main conclusion of this paper is that ‘multi-omics profiling enables individualized DSRCT treatment’, which is expected and a rather generic conclusion.

While the paper is well-written and easy to follow, the main concerns stem from the facts that the paper is rather descriptive in nature, rather appearing like a clinical case series, and that the main conclusions are obvious.

Response: We thank the reviewer for their careful reading of our manuscript and for acknowledging that it is well-written and easy to follow. While we acknowledge that our study involves the presentation of real-world data from a consecutive cohort of patients, we respectfully disagree with the assessment that the conclusions are self-evident or that the cohort is “relatively small.” With an incidence of approximately 0.2 per million persons per year, DSRCT is an ultra-rare cancer, and to our knowledge, no comparably large cohort of patients with this disease has previously been subjected to such comprehensive, multi-layered molecular profiling. In this context, a prospective series of 30 patients represents a unique and clinically meaningful dataset. As stated in the title and throughout the manuscript, this work is based on a structured precision oncology registry specifically designed to evaluate the clinical utility of multi-dimensional molecular profiling in patient care. By nature, such studies are descriptive in format, as they integrate high-dimensional clinical and molecular data to inform diagnosis and guide therapy in real-world settings. At the same time, they provide novel insights into recurrent biological features and potential therapeutic vulnerabilities that may subsequently be evaluated in dedicated clinical trials. These goals are distinct from, yet complementary to, mechanistic laboratory investigations, which, while valuable, rely on model systems that cannot fully capture the complexity of human disease.

In this context, we believe our study makes several novel and clinically relevant contributions. For example, by integrating phosphoproteomic data with transcriptomic and proteomic profiling, we show that ERBB2 is frequently expressed in DSRCT but not constitutively activated. This distinction is both biologically and clinically significant, as it argues against the use of therapies that depend on aberrant ERBB2 signaling activity, while supporting the rationale for next-generation ERBB2-directed antibody-drug conjugates such as trastuzumab deruxtecan (T-DXd),

which act independently of receptor activation. Our observation that DSRCT patients can experience durable responses to T-DXd provides, to our knowledge, the first patient-level evidence with extended follow-up supporting this therapeutic strategy in this ultra-rare disease. These findings have immediate implications for therapeutic decision-making and the design of clinical trials. Beyond ERBB2, our analyses uncovered additional actionable targets, including somatostatin receptor family members (SSTR3/5) and CLDN6. These findings are novel, have direct clinical utility by enabling access to early-phase clinical trials or off-label therapies, and open new therapeutic opportunities in a cancer that otherwise lacks effective treatment options and is associated with a dismal prognosis. Although not mentioned by the reviewer, these discoveries represent an important advance and illustrate the broader value of comprehensive multi-omics profiling in DSRCT.

With regard to the manuscript's conclusions, we respectfully note that the benefit of multi-omics analyses in this setting is not yet established and cannot be assumed. Indeed, this uncertainty is precisely why prospective registry programs such as MASTER exist, and why we are now pursuing additional, large-scale clinical trials to formally evaluate the impact of multi-layered profiling on patient outcomes. In response to the reviewer's comment, we have refined the wording of our conclusions to provide a more detailed reflection of our results. We hope these points clarify the intent, design, and relevance of our study, and we trust that the readers will appreciate the value of these findings within the appropriate clinical and translational framework. We further elaborate on these points in our responses to the specific comments below.

The fact that ERBB2 overexpression can be used as a predictive marker and target for therapy for cancer in general is well-established across cancer entities and not particularly novel for DSRCT. Several earlier reports have demonstrated this for DSRCT: 'HER2 Antibody-Drug Conjugates Are Active against Desmoplastic Small Round Cell Tumor', Zhang et al. 2024 Clin Can Res; 'High expression level of ERBB2 and efficacy of trastuzumab deruxtecan in desmoplastic small round cell tumour: a monocentric case series report', Brahmi et al. 2025 ESMO Open; 'Off-label use of fam-trastuzumab deruxtecan-nxki with early activity in a cohort of patients with desmoplastic small round cell tumor', Slotkin et al. 2024 ASCO Annual Meeting. These studies were based on even earlier work from Wu et al. (Multi-site desmoplastic small round cell tumors are genetically related and immune-cold, 2022 NPJ Precision Oncology), who proposed ERBB2 as a suitable target for a subset of DSRCT patients.

Response: We thank the reviewer for their comment and the opportunity to clarify the novelty and contributions of our work. We respectfully disagree with the suggestion that the role of ERBB2 as a therapeutic target in DSRCT is already well-established or that our study does not provide significant new insights beyond prior work.

At the time of manuscript preparation, only preclinical studies had explored ERBB2 as a potential target in DSRCT. We explicitly referenced and discussed these, including the important work by Zhang et al. (*Clin Cancer Res* 2024, PMID: 39120576), which demonstrated the efficacy of ERBB2-targeted agents such as T-DXd in xenograft models. However, to our knowledge, no peer-reviewed clinical study had validated these findings in patients at the time of submission. The case series by Brahmi et al. (*ESMO Open* 2025, PMID: 39921935) was published contemporaneously with our manuscript submission in February 2025 and was therefore not available for consideration

in the original version. As the reviewer notes, it supports the clinical potential of T-DXd in DSRCT and is now cited and discussed in the revised manuscript. That said, the Brahmi study is limited by a much shorter follow-up and does not provide an analysis distinguishing between ERBB2 expression and ERBB2 activation, a crucial factor in therapeutic decision-making. A key novelty of our work lies in addressing this distinction. Using phosphoproteomic analysis, we demonstrate that while ERBB2 is consistently expressed in DSRCT, it is not constitutively activated. This mechanistic insight, which was not conclusively addressed in prior studies, has immediate clinical implications: it argues against the use of drugs whose efficacy depends on aberrant ERBB2 signaling activity, such as afatinib, neratinib, or pertuzumab, and instead supports the rationale for next-generation antibody-drug conjugates like T-DXd, whose activity is independent of receptor activation. Our observation that DSRCT patients can achieve durable responses to T-DXd thus represents not only an important clinical validation but also a mechanistically informed approach to therapy selection in this ultra-rare cancer.

Regarding the study by Wu et al. (*NPJ Precision Oncology* 2022, PMID: 35379887), we thank the reviewer for raising it. This work, which we have cited in our original manuscript, reported RNA-level expression of *ERBB2* in DSRCT but did not assess receptor phosphorylation, signaling activity, treatment response, or the relative suitability of different ERBB2-targeted agents. While supportive of the idea that ERBB2 may play a biological role in DSRCT, it does not diminish the novelty of our findings, which provide patient-level clinical evidence that biomarker-guided selection of ERBB2-directed therapy can improve outcomes in this ultra-rare sarcoma.

Regarding citing practices, we have consistently referenced peer-reviewed, full-length publications. We did not cite unpublished data, conference abstracts, or meeting presentations – including our own (Renner et al., *Annals of Oncology* 2023 [ESMO Congress], <https://doi.org/10.1016/j.annonc.2023.09.1165>), which was the first to report on T-DXd treatment in DSRCT patients. We believe this approach is aligned with the standards of *Nature Communications* and ensures that cited material can be reliably evaluated by readers.

We hope this clarification helps situate our work appropriately within the existing literature and underscores its contribution to advancing precision oncology for DSRCT, a disease that currently lacks effective standard therapies.

Likewise, the possibility of treating DSRCT patients with pazopanib is not novel: ‘Pazopanib in advanced desmoplastic small round cell tumours: a multi-institutional experience’, Frezza et al. 2014 *Clinical Sarcoma Research*; ‘Clinical Activity of Pazopanib in Patients with Advanced Desmoplastic Small Round Cell Tumor’ Ludwig et al. 2018 *Oncologist*.

Response: We thank the reviewer for pointing out prior retrospective studies reporting the use of pazopanib in DSRCT. These are acknowledged in our manuscript, and we have cited both relevant publications (Frezza et al., *Clin Sarcoma Res* 2014, PMID: 25089183; Menegaz et al., *Oncologist* 2018, PMID: 29212731). However, only a minority of patients respond to pazopanib, and no predictive biomarkers have been established to guide its use. This key limitation – the inability to identify likely responders – has hindered the adoption of pazopanib as a standard therapy for DSRCT more than a decade after its initial introduction. In clinical practice, empirical use in refractory cases is common, while intensive multi-agent chemotherapy remains the dominant

approach, despite its limited efficacy. Our study adds a novel dimension by linking pazopanib responses to potential predictive biomarkers. By integrating multi-omics profiling – including mRNA and protein expression of pazopanib’s kinase targets and, importantly, kinase activity as assessed via phosphoproteomics – we provide a rational framework for biomarker-guided, individualized use of pazopanib and potentially other tyrosine kinase inhibitors (TKIs). We believe this approach has the potential to enhance therapeutic precision, reduce unnecessary toxicity, and optimize resource allocation. In our view, these findings extend beyond previous reports and support the rationale for a prospective clinical trial evaluating the biomarker-selected use of TKIs in DSRCT, while acknowledging the challenges inherent in conducting such trials in ultra-rare cancers.

To this reviewer, the most interesting part of this paper is the availability of the multiomics data. Unfortunately, the link and accession code provided to not give access to the data (yet).

Response: We appreciate the reviewer’s interest in the multi-omics dataset generated in this study, which we believe represents a resource of unprecedented scope that will hopefully inform future fundamental, translational, and clinical DSRCT research. All data have been deposited in the European Genome-Phenome Archive (EGA) under accession number EGAS00001007934. In accordance with EGA policy, the dataset will be made publicly available upon acceptance of the manuscript. To facilitate access during the review process, we have now obtained the following reviewer access tokens:

Whole-exome sequencing: <https://ega-archive.org/datasets/EGAD50000000910>

Whole-genome sequencing: <https://ega-archive.org/datasets/EGAD50000000911>

RNA sequencing: <https://ega-archive.org/datasets/EGAD50000000912>

DNA methylation profiling: <https://wwwdev.ebi.ac.uk/ega/studies/EGAS00001007934>

Another valuable point of this paper is that it is intriguing that 8 of 30 patients were misdiagnosed initially, which underscores that DSRCT is a diagnostic challenge and that new diagnostic biomarkers are required. In this context, the authors should include in the discussion section apart their appraisal of multiomics also other new diagnostic markers, such as CACNA2D2.

Response: Thank you for highlighting the importance of improving diagnostic accuracy in DSRCT and for drawing attention to CACNA2D2 expression, which was proposed as a new diagnostic marker during the review period of our manuscript (Geyer et al., *Cancer Commun* 2025, PMID: 40088092). As suggested, we have incorporated this recent finding into the discussion to complement our appraisal of the diagnostic value of multi-omics profiling.

Further, the authors should investigate the functional mechanism underlying the overexpression of ERBB2 and other proposed targets, such as SSTR3/5 and CLDN6, as well as their functional role(s) for the malignant phenotype of DSRCT. Otherwise, this descriptive report does not meet the bar for publication in such a prestigious journal, and appears to be more suitable for a specialized/clinical journal.

Response: We thank the reviewer for raising the question of mechanistic insight into the regulation and functional role of therapeutic targets in DSRCT. We fully agree that understanding the upstream drivers and downstream consequences of target expression is scientifically valuable. However, we respectfully submit that such questions lie far beyond any possible scope of the present study.

As clearly stated in the manuscript, this work represents a clinically focused analysis conducted within the framework of a nationwide precision oncology program, with the goal of informing diagnosis and guiding therapy in a real-world setting. It is not a laboratory-based, mechanistic investigation. The study design is necessarily descriptive in nature, as is standard and appropriate for precision oncology registry efforts. Our contribution lies in the integration of comprehensive multi-omics profiling with structured molecular tumor board decision-making, individualized experimental treatment, and systematic follow-up. Such clinically oriented studies generate first-hand evidence of how biomarker-guided therapies can translate into patient benefit in rare cancers where conventional clinical trials are challenging to implement. For this reason, similar precision oncology studies are widely recognized as valuable components of translational research and are regularly published in leading general-interest journals.

We certainly agree that functional and mechanistic studies are complementary to clinical research of this kind. However, it is important to note that even for ERBB2 – one of the most extensively studied oncogenes – the combined transcriptional, post-transcriptional, and protein-level mechanisms driving aberrant expression in the absence of gene amplification or activating mutations remain incompletely understood, despite decades of research in breast, gastric, gastroesophageal, and lung cancers (see, e.g., Cheng. *Cancers* 2024, PMID: 39062682). Expecting such mechanistic questions to be resolved in the context of our clinically anchored registry study is therefore not realistic. Extending this expectation to other proposed targets, such as SSTR3/5 or CLDN6, for which even less is known about upstream regulation and oncogenic function, would require multiple independent laboratory programs with entirely different aims, designs, and infrastructure.

We believe it is most appropriate to view our work and mechanistic investigations as complementary rather than overlapping. Our study was explicitly designed to generate actionable diagnostic and therapeutic insights in an ultra-rare, highly aggressive cancer by applying multi-omics profiling in a prospective, real-world setting. To our knowledge, no other study has assembled a DSRCT dataset of comparable breadth or clinical integration. In doing so, our work provides novel patient-level evidence that, in our view, justifies publication in *Nature Communications*. At the same time, we fully agree that detailed mechanistic studies merit separate, dedicated laboratory efforts. Importantly, the dataset we present offers a valuable foundation for such future work, including investigations into the biology and functional roles of ERBB2, SSTR3/5, and CLDN6 in DSRCT. We are confident that the revised manuscript effectively conveys both the scope and significance of our contribution.

Reviewer #3

The authors report new findings from the MASTER programme conducted by the German Cancer Research Center (DKFZ) in Heidelberg, focused on using multi-omics profiling to enable

personalised treatment recommendations for rare, aggressive cancers (DSRCT). This study expanded on previous omics-based approaches and combined genome/exome sequencing, RNA sequencing and DNA methylation profiling and proteomics and phosphoroteomics. This approach is a first in its breadth of components it combined, and found promising signals that a multi-layered molecular profiling can both correct wrong or incomplete diagnoses and provide effective personalised to slow or halt disease progression. These findings are of great interest to the oncological as well as the wider scientific community , as currently no effective standardised treatment regime for DSRCT exists, and a significant percentage of patients are misdiagnosed or receive a delayed diagnosis. As Nature Communications aims at a broader readership in contrast to purely clinical journals, revising some of the language and detailing both the methodology and the implications of the findings may improve accessibility.

Response: We thank the reviewer for their thoughtful and encouraging assessment of our work and for providing a clear summary of its scientific and clinical significance. We are grateful for the recognition of the novelty and potential impact of our multi-omics approach, particularly its potential to improve diagnostic accuracy and inform personalized treatment in DSRCT. We also appreciate the comment regarding accessibility for the broad readership of *Nature Communications*. Guided by the very helpful suggestions provided in the reviewer’s annotated version of the manuscript, we have clarified complex methodological aspects and refined the language to improve clarity for non-specialist readers.

This was an exploratory study, which was not powered appropriately to test for treatment superiority, therefore the authors make no claims of statistical significance. The overall study conduct was appropriate for the research objectives, however I have noted several questions regarding study specifics, timeline and recruitment criteria.

No statistical analysis was conducted, as this was not a powered superiority study. The authors report findings suitably with summary statistics, proportions and confidence intervals were appropriate. The data created for this study has been reported in full and is also available online. Full personalised treatment recommendations, DNA, RNA analyses etc. are available as supplementary information. There have been a few small clerical errors I referenced in the commented manuscript.

Response: We are very grateful to the reviewer for providing their biostatistical expertise and for the careful assessment of our study design and reporting. We also appreciate the thoughtful questions and comments provided in the annotated version of the manuscript, which we have addressed in the revision to improve clarity and transparency for the broad readership of *Nature Communications*. Thank you as well for noting the clerical errors; these have been corrected in the revised manuscript.

I share the authors’ conclusion, that the study provides evidence for both diagnostic and therapeutic improvements over the current standard of care. The authors have backed up their conclusion with specific numbers on how much of the treatment recommendations was attributable to the additional tests and screening. There is considerable reference to other studies, which is important to compare and give background to the study’s findings. However, the detail given is somewhat diluting the

novel results of the study. Perhaps some of the detail reported on the other studies can be shortened to keep the discussion section more focussed.

Response: We are pleased that the reviewer agrees with our main conclusion that multi-layered molecular profiling enables improved, individualized clinical management of DSRCT, a diagnostically challenging and clinically aggressive disease that continues to be treated with conventional chemotherapy despite very poor outcomes. We appreciate the suggestion to streamline references to prior studies, thereby keeping the discussion tightly focused on the novel contributions of our work. At the same time, Reviewers #2 and #4 encouraged a comprehensive integration of existing literature to ensure that our findings are properly contextualized. In light of these differing perspectives, we sought to find a balanced compromise: we retained detailed references where they are essential for context but also refined our text to keep the emphasis on the new insights generated by our study. We hope the reviewer will understand this challenge and agree with our chosen strategy. Of course, we would be happy to further shorten or streamline our discussion of prior studies if this were the editorial preference.

Relevant references have been used effectively throughout the manuscript to support statements and give background. In a few instances I have left comments asking for additional references.

As a biostatistician I am not qualified to assess the suitability of methods used for the sample processing and DNA, RNA and genome sequencing, and have therefore not commented on these aspects.

No pre-registration of the study protocol was reported, therefore no check of protocol deviations was performed. Ethics have been obtained prior to study beginning from the appropriate authorities, and was reported in the manuscript.

Suggestions on minor improvements are attached in the commented version of the manuscript.

Response: Thank you very much for your thoughtful suggestions regarding additional references and improvements to the wording of specific passages. We have carefully considered and incorporated these changes in the revised manuscript. We also appreciate your transparency regarding the scope of your expertise.

I would recommend accepting this manuscript subject to minor revisions.

Response: Thank you very much for your positive assessment and recommendation to accept the manuscript pending minor revisions. We have carefully addressed all remaining comments and updated the manuscript accordingly.

Reviewer #4

The manuscript by Renner et al. describes a multi-omics precision oncology approach for 30 patients with advanced desmoplastic small round cell tumor (DSRCT) enrolled in the German MASTER trial. Each patient's tumor underwent whole-genome/exome sequencing, RNA

sequencing, DNA methylation, and proteomic/phosphoproteomic profiling, with results discussed by a molecular tumor board (MTB) to recommend personalized therapies. In 93% of patients (28/30), the MTB was able to issue at least one therapeutic recommendation (median 3–4 per patient). In total, 107 individual recommendations were made across various categories including tyrosine kinase inhibitors (45% of recommendations), DNA damage repair (12%), immunotherapy (11%), theranostic approaches (10%), and others. Notably, 73% of all recommendations were based on RNA expression findings – for example, overexpression of certain tyrosine kinase (TK) pathway genes or cell-surface targets. The authors highlight that SSTR3/SSTR5 (somatostatin receptors) were over-expressed in about one-third of cases and CLDN6 in ~20% of cases, leading the MTB to frequently suggest peptide receptor radionuclide therapy (PRRT) or CLDN6-directed CAR T-cell therapy, respectively. Another key finding is that some DSRCT tumors showed elevated ERBB2 (HER2) mRNA expression without gene amplification, which prompted the MTB to recommend the HER2-targeted antibody–drug conjugate trastuzumab deruxtecan (T-DXd). Two patients who received off-label T-DXd experienced durable partial remissions (>12–15 months disease control), an impressive clinical outcome in this refractory sarcoma. The median overall survival from the first MTB consultation was reported as ~2.1 years, with a 4-year survival rate of ~10%, reflecting the aggressive nature of DSRCT despite heavy pretreatment. Overall, the study demonstrates that multi-layered molecular profiling can uncover actionable alterations in nearly every DSRCT patient, and that implementing MTB-recommended targeted therapies (such as multitarget TK inhibitors and novel ADCs) can lead to meaningful clinical responses in this ultra-rare, hard-to-treat sarcoma.

Response: We thank the reviewer for their thoughtful summary of our work and for acknowledging the impact of multi-layered molecular profiling in guiding biomarker-driven precision therapy for DSRCT. We are encouraged by this positive assessment and grateful for the recognition of the potential of our findings to improve outcomes for patients with this largely intractable disease.

Major Comments

1. Clarify MTB Decision-Making Rationale: The manuscript would benefit from a clearer explanation of how the Molecular Tumor Board formulated its recommendations in each case. Currently, the results enumerate the number and types of recommendations, but the rationale behind each recommendation is not sufficiently described. For example, it is unclear if the MTB followed predefined actionability criteria (e.g. ESMO Scale of Clinical Actionability of molecular Targets) or a structured algorithm. The authors should expand on their decision-making process: what specific molecular findings (alone or in combination) were deemed actionable, and why? Were recommendations based on levels of evidence from literature or on internal consensus/expert opinion? Detailing a few representative case vignettes could be helpful – for instance, why a particular gene’s overexpression or mutation led to a therapy recommendation, and how the MTB weighed other factors (prior therapies, patient condition, etc.) in that decision. Providing this information will make the process more transparent and reproducible. As an example, the text notes that 57% of patients were referred to clinical trials and that certain recommendations (e.g. PARP inhibition, immunotherapy) were sometimes made based on mutational signatures or expression alone; the manuscript should explicitly justify these choices. In summary, how did the MTB translate multi-omic data into a treatment plan? A brief methodological description (possibly

referencing Supplementary Fig. 2a) of the MTB workflow and decision criteria would greatly strengthen the paper's clinical relevance.

Response: We thank the reviewer for this insightful comment and for emphasizing the importance of transparency and reproducibility in the clinical decision-making process of the molecular tumor board (MTB). We fully agree that a clear understanding of how multi-omics findings are translated into therapeutic recommendations is essential for readers to appreciate the clinical relevance of our study. All recommendations made by the MTB are based on a standardized, structured workflow that includes a four-tiered system for evidence-based grading of molecular alterations. This system, developed by our consortium, has been widely adopted in Germany by networks such as the German Network for Personalized Medicine and the National Network for Genomic Medicine. It enables consistent and rigorous classification of molecular findings according to clinical actionability. The MTB's decisions are data-driven and rely on the integration of multiple parameters, including the type and level of evidence associated with a given alteration (e.g., FDA/EMA approval, clinical trial data, or preclinical evidence), prior therapies, tumor histology, and individual patient factors. This process is supported by extensive manual and automated annotation pipelines and has been described in detail in several published studies: Leichsenring et al., *Int J Cancer* 2019, PMID: 31008532; Horak et al., *Cancer Discov* 2021, PMID: 34112699; Horak et al., *Genes Chromosomes Cancer* 2022, PMID: 34331337; Mock et al., *NPJ Precis Oncol* 2023, PMID: 37884744. We believe that, to maintain the focus and clarity of the current manuscript, which centers on novel findings in DSRCT, it is most appropriate to direct readers to this established and peer-reviewed methodology. At the same time, we fully appreciate the reviewer's suggestion and have made several modifications to improve clarity: We now provide a more explicit summary of the MTB's decision-making framework in the revised Methods section (see new paragraph added). We include references to the above publications to guide interested readers to a detailed description of the evidence grading and interpretation process. We hope the reviewer will find this approach appropriate and sufficient to provide clarity on how molecular findings were translated into clinical recommendations, while allowing us to retain the focus of the manuscript on the novel clinical and molecular insights gained in DSRCT.

2. Include Survival Analyses for MTB Impact: I suggest an important analysis currently missing: a comparison of clinical outcomes between patients who received MTB recommendations versus those who did not, and between patients who followed an MTB-recommended therapy versus those who did not. Such an analysis would address the utility of the MTB's guidance. For instance, only 28 of 30 patients received recommendations (two patients had none) – how did their outcomes compare? More critically, among the 28 with recommendations, only 13 patients actually received an MTB-recommended molecularly guided therapy (as per Table 2). Do these 13 patients exhibit any trend toward improved survival (or longer progression-free survival) compared to those who, for whatever reason, did not receive a recommended therapy? Even acknowledging selection biases (patients able to receive these therapies might have better performance status, etc.), presenting a Kaplan–Meier analysis or summary statistics for overall survival in these subgroups would be very informative. For example, one could compare median OS from MTB enrollment for “MTB-treated” patients vs. “non-MTB-treated” patients. If no significant difference exists, that is also worth discussing candidly. This comparative survival analysis would directly address the clinical impact of the MTB program – a key question for a reader. The lack of such analysis is noted by the reviewer, and adding it (even if descriptive due to small numbers) would strengthen

the manuscript's claims that the MTB's recommendations confer meaningful benefit. If numbers are too small for formal statistics, the authors can still qualitatively comment on any survival differences. (For instance, it appears that patients who accessed recommended targeted therapies achieved some prolonged disease control as noted in specific cases, but a cohort-level comparison is needed to generalize this.)

Response: We appreciate the reviewer's valuable and constructive suggestion. We fully agree that assessing clinical outcomes in relation to MTB guidance is central to evaluating the utility of precision oncology programs. In principle, comparisons between patients who received and implemented MTB recommendations versus those who did not would be highly informative. However, as is common in registry-based studies, follow-up data are often incomplete for patients who did not proceed with MTB-guided therapies, limiting our ability to perform reliable between-group analyses in this ultra-rare disease. Indeed, another reviewer with statistical expertise (Reviewer #3) emphasized that formal comparative survival analyses would not be appropriate given the sample size and inherent selection biases.

To address the reviewer's request within these constraints, we performed an intra-patient progression-free survival ratio (PFSr) analysis, as introduced by von Hoff et al. (*J Clin Oncol* 2010, PMID: 20921468). The PFSr is defined as the duration of PFS on molecularly guided therapy divided by the duration of PFS on the immediately preceding systemic therapy. This approach allows each patient to serve as their own control, thereby reducing confounding from differences in performance status or access to treatment. According to widely accepted expert consensus, a PFSr greater than 1.3, and in some studies greater than 1.5, is considered indicative of clinical benefit.

In our cohort, seven of 13 patients (53.8%) who received MTB-recommended therapies achieved a PFSr greater than 1.3. This proportion compares favorably with landmark precision oncology trials, including MOSCATO 01 (33%; Massard et al., *Cancer Discov* 2017, PMID:28365644) and WINTHER (25%; Rodon et al., *Nat Med* 2019, PMID: 31011205), as well as a recent global analysis of MASTER (35.7%; Horak et al., *Cancer Discov* 2021, PMID: 34112699), and lies above the range reported in a recent meta-analysis of precision oncology studies (33–44%; pooled estimate, ~38%; Gladstone et al., *NPJ Precis Oncol* 2025, PMID: 40175535). Although the number of patients in our cohort precludes definitive statistical comparisons, we believe that the favorable PFSr distribution in our study adds meaningful support for the value of MTB guidance in DSRCT. At the same time, we caution that direct cross-trial comparisons should be interpreted carefully, given differences in patient populations, trial designs, and therapeutic landscapes.

We have added these analyses and a corresponding discussion to the revised manuscript, including a graphical summary of the PFSr distribution, to more explicitly convey the clinical impact of MTB-guided therapy in this cohort.

3. Define Expression Thresholds for “Overexpression”: A major point of confusion is how the study defined a gene as “overexpressed” or having “increased mRNA expression.” Many of the MTB's recommendations – 78 of 107 (73%) – were based on elevated transcript levels of potential targets. However, the manuscript does not specify what thresholds or reference comparisons were used to call something “overexpressed.” The authors should clearly state the criteria for deeming

a target's mRNA upregulated. For example, did they compare the tumor's transcripts per million (TPM) values against a reference dataset of other sarcomas or normal tissues? Was a specific fold-change or percentile rank used as a cutoff? The absence of such definitions leaves the reader wondering how targets like ERBB2, CLDN6, SSTR3, or SLFN11 were selected for action. The Supplementary Figures hint that the authors compared expression in DSRCT relative to other sarcoma types (e.g. Supplementary Fig. 4 shows SSTR family expression in DSRCT vs other sarcomas) – indeed, SSTR3 and SSTR5 were consistently higher in DSRCT than in other sarcomas. If this comparative approach was used generally, it should be stated. Similarly, the manuscript notes that the MTB recommended HER2-targeted therapy (T-DXd) in seven patients due to increased ERBB2 mRNA, and recommended CLDN6-targeted CAR-T in some patients due to CLDN6 expression. The exact cutoff (if any) for “high” ERBB2 or CLDN6 expression should be explained. Was any detectable CLDN6 in RNA-seq considered actionable (given CLDN6 is normally absent in adult tissues), or did it have to exceed a TPM threshold? For HER2, did the authors have an expression level equivalent to “HER2-low” or “HER2-high” as per known breast cancer criteria, or was it purely relative within their cohort? Likewise, SLFN11 expression: the text references “high SLFN11” in one patient prompting a PARP inhibitor recommendation, but how was “high” defined? Clarity on these points is crucial, because the concept of actionability based on RNA is relatively new – readers need to know how robustly overexpression was determined. The authors should also discuss the rationale for considering mere mRNA overexpression as sufficient to target, especially for cases like ERBB2 where no gene amplification or mutation was present (i.e. why presume that high transcript alone makes the protein a good target?). Overall, please add a Methods description or supplementary table detailing the expression-based target calling strategy (thresholds, comparisons, and any validation by protein IHC if done). This will greatly enhance the transparency and reproducibility of the study's approach.

Response: We thank the reviewer for this important and constructive comment. We fully agree that clarity in defining expression thresholds is essential, especially since a substantial proportion of our MTB recommendations were based on transcriptomic findings. We are grateful for the opportunity to elaborate on how we defined “overexpression” at both the RNA and protein levels, how these criteria were applied across all potential targets, and why, in specific cases, expression alone provided sufficient rationale for clinical action.

As a general principle in the MASTER program, expression, whether at the transcript or protein level, is always evaluated relative to a reference cohort. For RNA, genes listed in the *Cancer Gene Census* (<https://www.sanger.ac.uk/data/cancer-gene-census>) as well as in our in-house lists of druggable genes and genes of interest, were considered overexpressed if their fold change exceeded two times the median of a reference cohort of 148 RNA-sequenced samples. Since October 2017, an additional rule has been applied: for genes never found to be upregulated by a factor of 2, a z-score greater than 2 was also considered indicative of overexpression. These standardized criteria were applied systematically to all potential targets across all patients, ensuring consistency and reproducibility of the results. For proteins, the same principle of relative comparison was applied against a sarcoma reference cohort. For example, the ERBB2 protein expression value cited (log₁₀ intensity difference of 0.67) corresponds to a high expression level, with a mean log₁₀ intensity for DSRCT of 8.01 compared to 7.33 in the pan-sarcoma reference cohort. Recognizing that log-transformed fold changes are not always intuitive, we also present these data on the linear scale, which yields an approximate fold change of 4.7. In proteomic

analyses, a fold change greater than 2 (or less than 0.5 for downregulation) is commonly accepted as a meaningful threshold. Statistical significance was assessed using Welch's t-test to account for unequal sample sizes. For large-scale differential analyses (e.g., Figure 4c), Benjamini-Hochberg correction was applied to control the false discovery rate, allowing us to identify proteins with both statistical and biological relevance.

Importantly, no target was selected irrespective of our defined overexpression criteria. However, as the reviewer correctly points out, for certain targets, even lower levels of expression may be clinically meaningful. For instance, CLDN6 is absent from normal adult tissues, so any detectable expression may be actionable, particularly in the context of emerging T-cell therapies. For ERBB2, accumulating evidence from breast and gastric cancers demonstrates that even low or ultra-low expression can be effectively targeted by next-generation antibody-drug conjugates, such as trastuzumab deruxtecan (T-DXd), whose efficacy does not depend on high target abundance or kinase activity.

Regarding the reviewer's question on how our criteria relate to the ERBB2-high or ERBB2-low categories defined in breast cancer, these categories are based on immunohistochemistry (IHC), which was not systematically performed in our study. Accordingly, our RNA sequencing and proteomic data cannot be mapped precisely onto these clinical scoring systems. Nevertheless, in one patient with a durable response to T-DXd, IHC was available and showed an ERBB2-low staining pattern, consistent with actionability despite the absence of amplification or strong IHC positivity. This supports the idea that ERBB2-low expression is clinically relevant in DSRCT, although further studies will be required to determine whether even lower levels of expression are actionable in this disease. A central novelty of our study is that we identify increased ERBB2 expression in DSRCT in the absence of amplification, activating mutations, or, most importantly, pathway hyperactivation. This distinction was made possible by applying mass spectrometry-based phosphoproteomics, a technique that had not previously been applied to this sarcoma type. The resulting biological insight has direct therapeutic implications: it supports the use of T-DXd, whose mechanism of action is independent of ERBB2 signaling, while arguing against drugs such as trastuzumab, pertuzumab, afatinib, or neratinib, which require receptor activity and/or dimerization and had been previously suggested as treatment options for DSRCT. Thus, it is not mere transcript abundance that informs our recommendations, but rather the integration of expression data with pathway activity profiling to select the mechanistically most appropriate drug.

Finally, we would like to thank the reviewer for recognizing the novelty of our approach to nominate therapeutic targets based on expression data. We emphasize that our study was not designed to determine absolute quantitative threshold values for new therapeutic targets; this important question will require further investigation in larger patient cohorts and across various treatment contexts. Nevertheless, for some targets such as CLDN6 or ERBB2, the clinical evidence already suggests that even low levels of expression may be actionable.

In response to the reviewer's comment, we have revised the Methods section and figure legends to provide explicit descriptions of the criteria, reference cohorts, and statistical approaches used, and we now present protein expression differences in both log and linear scales for improved interpretability. We hope these clarifications will make our approach more transparent and reproducible, while also underlining the rationale and novelty of using multi-omics expression data to inform therapy decisions in this rare and otherwise treatment-refractory cancer.

4. Provide Literature Context for SLFN11 and PARP Inhibitor: The recommendation of PARP inhibitor therapy in SLFN11-positive patients is biologically plausible, but the manuscript needs to better justify this with prior research. The authors note that one patient had increased SLFN11 expression (along with mutational signature SBS8) which led the MTB to advise a PARP inhibitor (olaparib) combined with trabectedin. They also mention up to 10 patients had recommendations for PARP inhibitors due to HR pathway deficiencies (SBS3/8 or BRCA-pathway mutations) and in some cases high SLFN11. However, the rationale for using SLFN11 as a biomarker of PARP inhibitor sensitivity should be explicitly stated. Prior studies (e.g. in Ewing sarcoma, small cell lung cancer, etc.) have shown that SLFN11 expression correlates with responsiveness to DNA-damaging agents and PARP inhibitors. If available, the authors should cite relevant literature to support this approach. For instance, it is known that SLFN11 interferes with replication fork recovery, making tumors with high SLFN11 more vulnerable to agents like platinum or PARP blockade. By including a sentence in the Discussion such as, “This recommendation was motivated by reports that SLFN11-high tumors are especially sensitive to PARP inhibition ,” the authors can ground their MTB decision in published evidence. Additionally, if the patient who received olaparib (DSRCT-14 in Table 2) achieved stable disease for 4 months, that outcome could be contextualized by noting it aligns with the hypothesized role of SLFN11. In summary, please add a brief discussion of why SLFN11-positive status justified PARP inhibitor therapy, with references to prior studies of SLFN11 as a predictive biomarker. This will help readers understand the biological reasoning and novelty of applying that rationale in DSRCT.

Response: We thank the reviewer for this constructive comment and fully agree that providing a literature context for our recommendation of PARP inhibition in SLFN11-positive DSRCT patients is important.

As the reviewer points out, a substantial body of evidence indicates that SLFN11 expression is a predictive biomarker of sensitivity to DNA-damaging agents, including PARP inhibitors. Mechanistically, SLFN11, a DNA/RNA helicase, blocks the recovery of stalled replication forks, thereby enhancing tumor cell death in response to PARP inhibition or platinum-based chemotherapy (Coleman et al., *Br J Cancer* 2021, PMID: 33328609; Winkler et al., *Br J Cancer* 2021, PMID: 33339894; Zoppoli et al., *Proc Natl Acad Sci U S A* 2012, PMID: 22927417). Across multiple cancer types, high SLFN11 expression has been associated with improved outcomes to PARP inhibitors, including in small-cell lung cancer (Lok et al., *Clin Cancer Res* 2017, PMID: 27440269; Pietanza et al., *J Clin Oncol* 2018, PMID: 29906251) and Ewing sarcoma (Tang et al., *Clin Cancer Res* 2015, PMID: 25779942). Moreover, SLFN11 is frequently expressed in pediatric sarcomas, including DSRCT, highlighting its potential role as a biomarker in this setting (Gartrell et al., *Mol Cancer Ther* 2021, PMID: 34413129).

In our study, one patient (DSRCT-14) with high SLFN11 expression and single-base substitution signature 8 (SBS8) was recommended treatment with olaparib in combination with trabectedin and achieved stable disease for four months. This clinical observation aligns with the hypothesized role of SLFN11 in mediating sensitivity to PARP inhibitors.

In line with the reviewer’s suggestion, we have expanded the revised manuscript to explicitly reference this body of evidence and to explain the rationale for nominating PARP inhibition in SLFN11-positive DSRCT patients.

5. Clarify Basis for Recommending Immunotherapy: The manuscript mentions that 12 of 107 recommendations (11%) fell into the “IT” (immunotherapy) basket, including suggestions such as PD-1 checkpoint blockade and CLDN6-directed CAR T-cells (the latter could also be considered cell therapy). However, it is unclear what criteria were used to recommend immune checkpoint inhibitors in this cohort. DSRCT is generally not known as an immunologically “hot” tumor, so on what basis did the MTB suggest PD-1 inhibitor therapy (e.g., nivolumab for patient DSRCT-03, per Table 2)? Did any patients have biomarkers like high tumor mutation burden, mismatch-repair deficiency, PD-L1 expression, or prominent T-cell infiltrates? The supplementary data (Table 6) intriguingly shows one patient (DSRCT-19) with mutational signature SBS2 who was given an “immune checkpoint” recommendation. SBS2 is an APOBEC-related signature; the connection between SBS2 and immunotherapy efficacy is not obvious. If the authors used any immune gene expression signatures (for example, a tertiary lymphoid structure [TLS] signature or cytotoxic T-cell signature from the RNA-seq) or any specific immunologic classifier to identify patients who might benefit from immunotherapy, this should be stated. The reviewer specifically asks whether TLS gene signatures were used – since TLS presence has been correlated with immunotherapy response in some sarcomas, did the authors evaluate this or other metrics (like the “immune hot/cold” expression clusters)? If not, the authors should clarify that immunotherapy was recommended on more general grounds (e.g., last-line option or extrapolation from other sarcomas) rather than on a clear predictive biomarker. In the revised manuscript, please explain the rationale for immunotherapy recommendations: was it based on any molecular/immunological evidence or simply a trial enrollment opportunity? For transparency, list what factors (if any) led to recommending PD-1/PD-L1 blockade in those cases. This will preempt readers’ questions about whether those suggestions were driven by data (for example, “Patient X had moderate CD8+ T cell infiltration and PD1 gene expression, justifying nivolumab trial”) or whether they were more empirical. Such clarification is important, given the current lack of approved immunotherapy indications in DSRCT.

Response: We thank the reviewer for this thoughtful and important comment. We appreciate the opportunity to clarify the rationale behind the immunotherapy recommendations made by the MTB, which encompass a range of modalities, including immune checkpoint inhibitors and cell-based therapies such as CLDN6-directed CAR T cells. As the reviewer rightly points out, many sarcomas, including DSRCT, are generally considered immunologically “cold” and less responsive to immunotherapy compared to other cancer types. We agree that clarifying the molecular basis for recommending immunotherapy in this setting is particularly important.

As part of our comprehensive multi-omics profiling approach, we routinely assess a broad spectrum of predictive biomarkers that extend beyond those captured by conventional panel-based methods. These range from single-gene alterations at the DNA and/or RNA level – such as *CD274* (*PD-L1*) amplification and overexpression or aberrant expression of targets relevant to cellular therapies like *CLDN6*, *PRAME*, and *MAGEA* family members (whose clinical utility is often further supported by simultaneous HLA genotyping from whole-genome data) – to complex biomarkers, including tumor mutational burden and mutational signatures.

In the revised manuscript, we provide a supplementary table that details the molecular features supporting each of the 12 immunotherapy recommendations. All recommendations were based on defined molecular alterations and were graded according to evidence levels established by our consortium and widely used in Germany (Leichsenring et al., *Int J Cancer* 2019, PMID: 31008532;

Horak et al., *Cancer Discov* 2021, PMID: 34112699; Horak et al., *Genes Chromosomes Cancer* 2022, PMID: 34331337; Mock et al., *NPJ Precis Oncol* 2023, PMID: 37884744). None were based on empirical considerations or trial availability alone.

Regarding the recommendation of immune checkpoint inhibition for a patient with an APOBEC-associated mutational signature, i.e., SBS2, this was based on emerging evidence that APOBEC-driven mutagenesis is associated with increased tumor immunogenicity and improved response to immune checkpoint inhibitors across multiple cancer types. Studies have shown that tumors harboring APOBEC signatures often exhibit increased neoantigen load, immune activation, and T-cell infiltration – even in tumors with otherwise low tumor mutational burden (Boichard et al., *Oncoimmunology* 2018, PMID: 30723579; Wang et al., *Oncogene* 2018, PMID: 29695832; Chen et al., *Cancer Sci* 2019, PMID: 31222843). In particular, SBS2 and related signatures have been linked to favorable responses to PD-1/PD-L1 blockade in entities such as breast and bladder cancer (DiMarco et al., *Cancer Immunol Res* 2022, PMID: 34795033; Lu et al., *Front Mol Biosci* 2025, PMID: 40356722; Shi et al., *Theranostics* 2022, PMID: 35673559; Boll et al., *Nat Commun* 2025, PMID: 39979258). These data support the potential role of APOBEC signatures as predictive biomarkers for immunotherapy benefit, which we considered in the MTB’s recommendation in this case.

We also thank the reviewer for raising the question of whether transcriptome-based immune signatures, such as those indicative of tertiary lymphoid structures or cytotoxic T-cell activity, were used. While we are actively working to integrate such complex RNA-based classifiers into our clinical decision-making framework, they are not yet part of the standardized workflow in the MASTER program. We believe that these metrics hold promise for further refining immunotherapy recommendations and are committed to advancing their clinical implementation.

6. Contextualize Novelty and Prior Literature on Targeted Therapies in DSRCT: The study’s findings should be better positioned within the context of existing literature, to highlight both consistency with known results and the novel contributions. The authors are investigating an ultra-rare sarcoma where little targeted therapy data exists, so any connections to prior studies should be emphasized.

Response: We fully agree that situating our findings within the context of prior research is essential to highlight both areas of consistency and the novel contributions of our study. In revising the manuscript, we faced somewhat divergent recommendations: Reviewers #2 and #4 encouraged us to provide a detailed integration of existing literature, while Reviewer #3 suggested shortening references to prior work to avoid diluting our own novel findings. As a compromise, we have placed greater emphasis on highlighting the new insights provided by our study. We hope that this strikes the right balance between the different perspectives. Of course, we would be happy to further adjust the level of detail on prior studies if the reviewers or editors feel that additional streamlining is warranted.

For instance, pazopanib (multi-kinase inhibitor) has been used off-label in DSRCT before – a retrospective series of 29 patients (Ludwig et al., 2018) showed disease stabilization in ~55% and a median PFS of ~5.6 months. In the manuscript’s Table 2, multiple patients received pazopanib

due to various TK gene overexpressions, with several achieving partial responses or stable disease lasting 6–17 months, which is quite encouraging and generally in line with the prior report. Citing such studies would show that the authors are building on a foundation (e.g., using pazopanib as a baseline known active drug in DSRCT) and then going further by personalizing its use to those with specific angiogenic/kinase signals.

Response: We thank the reviewer for recognizing the clinical relevance of our findings and for emphasizing the potential role of pazopanib in the treatment of DSRCT. We fully agree that our work builds upon important prior studies, including the retrospective series by Ludwig et al., which we have cited in the manuscript (Menegaz et al., 2018, *Oncologist*, PMID: 29212731). These earlier reports demonstrated encouraging responses to pazopanib but were not guided by predictive biomarkers, and the drug has not been established as a standard treatment option for DSRCT. Our study provides substantial novel insights by showing that comprehensive molecular profiling – including mRNA and protein expression of pazopanib-relevant tyrosine kinases, as well as direct measurement of kinase activity through phosphoproteomics – can inform a biomarker-guided, individualized use of this agent. We believe this mechanistically driven approach surpasses previous work by providing a rational framework for identifying likely responders, with tangible implications for enhancing clinical decision-making in this ultra-rare and challenging disease. We also note that a similar point was raised by Reviewer #2, and we have provided a detailed response to that comment. We are grateful that both reviewers independently recognized the importance of this aspect of our work.

The most striking novel finding here is the efficacy of HER2-targeted ADC (T-DXd) in DSRCT. This is, to my knowledge, the first formal report of clinical responses to an ADC in DSRCT patients, supported by the authors' multi-omics rationale. The Introduction references preclinical evidence that Her-family signaling could be a vulnerability in DSRCT, including a report that afatinib, cetuximab, and T-DXd reduced DSRCT xenograft growth. The authors should explicitly state that their two patient cases on T-DXd represent a translational validation of those preclinical findings. In fact, a very recent case series (likely emerging in 2024–2025) has also suggested that HER2 is a therapeutic target in DSRCT and that T-DXd can benefit these patients – the authors may want to reference this if available, as it underscores the relevance of their approach. Overall, the Discussion should be refined to stress the study's novelty: multi-layered molecular profiling in DSRCT has not been done at this scale before, and the results reveal previously unappreciated therapeutic options (e.g., PRRT for SSTR3-positive disease, CLDN6 CAR-T for CLDN6-expressing tumors, ADC for HER2-low tumors). At the same time, the findings are consistent with emerging paradigms in rare cancers, where RNA or proteomic data can identify actionable targets even in the absence of classic genomic alterations. Emphasizing this consistency and novelty will help convince readers (and the editors) that the manuscript is a significant advance worthy of publication.

Response: We sincerely thank the reviewer for the thoughtful and encouraging assessment of our work. We are particularly grateful for the recognition of the clinical significance and novelty of our finding that DSRCT patients benefit from treatment with trastuzumab deruxtecan (T-DXd). As the reviewer correctly notes, our results are consistent with earlier preclinical work showing that ERBB2 could represent a therapeutic vulnerability in DSRCT, and that T-DXd, among other agents, reduced tumor growth in DSRCT xenograft models (Smith et al., *Dis Model Mech* 2022,

PMID: 34841430; Zhang et al. *Clin Cancer Res* 2024, PMID: 39120576). We agree that our study provides important clinical validation of these findings and now explicitly highlight this connection in the revised manuscript.

In addition, we have cited the recently published case series on T-DXd in DSRCT (Brahmi et al., *ESMO Open* 2025, PMID: 39921935), which was published at the time of our manuscript submission in February 2025 and was therefore not available for consideration in the original version. As the reviewer suggests, this report further supports our observation that T-DXd may offer clinical benefit to DSRCT patients. Notably, the follow-up period in our study is substantially longer than in the study by Brahmi et al., allowing for a more reliable assessment of the durability of responses to T-DXd. However, we would like to note that our study extends beyond validation by introducing a more refined biomarker framework. Through the integration of phosphoproteomic analysis, we demonstrate for the first time that ERBB2 is *expressed* in DSRCT but not *aberrantly activated*. This distinction has important therapeutic implications. While previous studies (e.g., Smith et al., *Dis Model Mech* 2022, PMID: 34841430) have proposed ERBB2-targeted therapies such as afatinib or neratinib – agents that rely on inhibition of ERBB2’s catalytic activity – our data suggest that only antibody-drug conjugates like T-DXd, whose mechanism of action is independent of ERBB2 kinase activity, are likely to be effective in this setting. Accordingly, our findings suggest that kinase-dependent ERBB2-targeted therapies are not suitable for DSRCT, supporting the use of biomarker-guided selection of the most appropriate ERBB2-directed agent.

As recommended, we have revised the Discussion section to more explicitly highlight this novel insight and to emphasize the broader impact of our approach. Specifically, our study represents the first large-scale application of multi-layered molecular profiling in DSRCT and has uncovered a range of actionable targets – including SSTR3, CLDN6, and ERBB2 – that carry immediate clinical relevance. These findings not only validate but also extend prior knowledge and provide a rationale for personalized treatment in a cancer that currently lacks effective standard therapy.

We also fully agree with the reviewer’s important observation that transcriptomic and (phospho)proteomic profiling can reveal actionable biology even in mutation-poor cancers like DSRCT. We are particularly enthusiastic about the potential of this strategy to open up new therapeutic opportunities in rare sarcomas and other fusion-driven cancers in adolescents and young adults, where genomic profiling alone may fall short.

We hope the reviewer agrees that these revisions strengthen the manuscript and more clearly convey the novelty, consistency, and clinical implications of our work to both readers and editors.

Minor Comments

Gene Expression Cutoffs in Text: When describing expression-based findings, the manuscript currently uses qualitative terms like “overexpression” or “increased expression” without numeric detail. In revision, consider reporting the actual expression levels for key targets in a comparative sense. For example: “ERBB2 was expressed at 70 TPM in Patient 28’s tumor, which is $>4\times$ the median level in our DSRCT cohort. Similarly, when stating “consistent overexpression of SSTR3 and SSTR5 in DSRCT compared to other sarcomas”, it would help to quantify (“median TPM X

vs Y in other sarcomas”) either in text or a supplemental table. This would complement the needed clarification of thresholds (as per Major Comment #3).

Response: Thank you for this valuable and constructive comment. The importance of providing quantitative detail alongside qualitative descriptors was also noted by Reviewer #1. In the revised manuscript, we have specified the standardized criteria applied within the clinical framework of the MASTER registry to evaluate mRNA expression levels. We have also added a detailed explanation of how corresponding protein expression was assessed, emphasizing that median expression levels of key targets, such as SSTR3, SSTR5, and ERBB2, were systematically compared to those in a large reference cohort of non-DSRCT sarcomas. We believe these revisions substantially improve the clarity, transparency, and reproducibility of our analysis, making the rationale for prioritizing clinically actionable targets through integrative multi-omics profiling more accessible to the reader.

Terminology – “Constitutive signaling”: The phrase “ERBB2 overexpression in the absence of constitutive ERBB2 signaling” might be confusing to readers. It appears to mean that no activating mutation or high phosphorylation of ERBB2 was present (i.e., no evidence of strong signaling activity), yet the gene was highly transcribed. The authors might rephrase this for clarity. Perhaps specify that no ERBB2 kinase activation was detected via phosphoproteomics, supporting the use of an ADC (which requires surface expression but not signaling). This distinction between targeting a protein for its expression vs. targeting an active driver should be made clear to avoid misinterpretation.

Response: Thank you for highlighting this crucial point. As you correctly noted, the phrase “absence of constitutive ERBB2 signaling” refers to ERBB2 expression without aberrantly elevated activity. This distinction is critical, and we fully agree that it must be clearly communicated, given its biological significance and immediate clinical implications. To ensure that readers grasp this key message, we have carefully revised the wording and explicitly emphasized the difference between mere ERBB2 expression and its pathologic activation – an insight that advances our understanding of DSRCT biology and directly informs the selection of appropriate targeted therapies.

Proteomics Results – Integration: The paper mentions proteomic/phosphoproteomic profiling was performed in 9 patients, and gives a few examples (e.g., FGFR4, KDR phosphorylation in certain cases). However, it’s not entirely clear how these proteomic findings were used. The authors should ensure they close the loop on those results. For instance, patient DSRCT-18 had activated KDR and TYRO3 by phosphoproteomics and did respond to pazopanib (which hits VEGFR/KDR) for 17 months – this is a compelling connection that could be highlighted as a success of multi-omics integration. Conversely, the lack of ERBB2 phosphorylation in the two T-DXd-treated patients is noted as justification for choosing an ADC over a small-molecule inhibitor. These points do appear in the text, but the authors might make them more explicit so that the contribution of proteomics is appreciated. If proteomic data did not significantly change recommendations (beyond confirming known targets), that could be stated too, as it teaches where multi-omics adds value.

Response: Thank you for this insightful comment, which highlights a key strength of phosphoproteomic analysis: its ability to directly assess the functional state of signaling molecules – information that cannot be reliably inferred from genomic or transcriptomic data alone. The two examples mentioned clearly demonstrate the added value of this approach by illustrating its impact on two major classes of targeted cancer therapies. Small-molecule tyrosine kinase inhibitors, such as pazopanib, are effective only in tumors driven by aberrant activation of kinases within their target spectrum. Phosphoproteomic profiling thus enables a more rational and selective use of these agents by identifying patients most likely to benefit. Conversely, the absence of aberrant ERBB2 signaling in DSRCT argues against the use of therapies targeting its catalytic function, including small-molecule inhibitors and monoclonal antibodies such as trastuzumab and pertuzumab. In contrast, our data support the use of an antibody-drug conjugate, which exerts its cytotoxic effect independently of kinase activation and was associated with durable responses in extensively pretreated patients. As recommended, we have further emphasized these mechanistic and clinically actionable insights in the revised text to ensure their significance is clearly conveyed.

Formatting and Typographical Issues: Overall the manuscript is well-written. We caught a few minor formatting issues that should be corrected:

In Supplementary Table 3 (PDF), the percentage “12,1%” is written with a comma; this should be “12.1%” in English decimal format . Please ensure consistency in numeric formatting (use a period for decimals in the main text and supplements).

Response: Thank you for spotting this error. We inadvertently used German formatting in this instance, which has now been corrected.

There are instances of awkward line breaks in the PDF (e.g., the radioisotope fluorine-18 in 18F-FDG-PET appears split across lines in our copy). In the final production, such issues will likely be fixed by copyeditors, but the authors should double-check the rendering of all technical terms and units (especially in figure labels).

Response: Thank you for bringing this to our attention. We have carefully reviewed all technical terms and units to ensure their accuracy and consistency. We apologize for any artifacts that may have resulted from the PDF conversion during the submission process. We will work with the editorial office to ensure that all elements are presented correctly in the final version.

All acronyms should be defined at first use. For example, “MTT” is used in Figure 2 legend (for “molecularly targeted treatment”) – ensure this is defined in the text or figure caption so readers know it refers to treatments given outside trials based on MTB recommendations

Response: Thank you for this important comment. We have ensured that all abbreviations and acronyms are defined upon first use. Additionally, we have clarified in the legend to Figure 2 that “molecularly targeted treatment” (MTT) refers to therapies recommended by the MTB.

Gene symbols are appropriately capitalized; just verify if the journal prefers them italicized. In Nature journals, human gene symbols are typically written in roman (non-italic) uppercase, which seems to be what the authors have done (e.g., WT1, EWSR1 in text). This is fine, but ensure consistency (e.g., in one case “EWSR1::WT1” fusion was clearly described; make sure all gene fusion notations use the same format throughout).

Response: As suggested, we have carefully reviewed the entire manuscript, including all figures, tables, and supplementary data, to ensure that all gene symbols and fusion genes are accurately presented and fully comply with the formatting guidelines of *Nature Communications*.

Minor grammar: page 2 of the PDF has author affiliations where a name is split by a line break (“Annika BaudeMüller”) – this is just a PDF artifact, not an issue for content. No significant typos were noted in the main text.

Response: Thank you very much for taking the time to carefully read the entire manuscript. We would like to confirm that the author’s name is correctly spelled as *Annika Baude-Müller*, and we hope it will appear accurately in the PDF version of the revised manuscript.

Data Availability and Figures: The authors should double-check that all figures and tables are properly referenced in the text in order. For instance, Figure 5e is cited when discussing ERBB2 expression and IHC – make sure Figure 5e indeed shows what is described (ERBB2 mRNA and protein levels). Similarly, ensure that each supplementary figure/table is cited at the appropriate point. A quick check suggests everything is cited, but it’s worth verifying consistency one more time (e.g., Supplementary Figure 2a is mentioned as containing the multi-omic biomarker overview so Supplementary Fig. 2 should indeed illustrate that workflow).

Response: Thank you for this valuable comment. We have carefully re-examined all figure and table callouts and made the appropriate corrections.

Also, minor point: in Table 2, the footnotes A, B, C are very useful to explain combination treatments – ensure these footnotes are included in the main text or legend for clarity (they appear to be in the PDF bottom of Table 2).

Response: Thank you for this helpful suggestion. In the revised manuscript, the former Table 2 has been replaced by a new Figure 5, which provides comprehensive information on all biomarker-based therapies implemented. The information that was previously included in the footnotes of Table 2 has been integrated into the legend of Figure 5 to ensure greater clarity and transparency.

By addressing the above comments – especially the major points about methodology clarity and additional analyses – the authors can substantially improve the manuscript’s clarity and impact.

Response: We are grateful for your detailed review and numerous constructive suggestions, which have greatly enhanced the clarity, rigor, and overall quality of our manuscript.

NCOMMS-25-13803B

Multi-Layered Molecular Profiling Informs the Diagnosis and Targeted Therapy of Desmoplastic Small Round Cell Tumor

Thank you for the opportunity to revise our manuscript further. We have carefully addressed all remaining reviewer comments and provide a brief summary of the changes below.

Reviewer #1

The overlapping labels for DSRCT and MPNST in Figure 4a have been corrected. No additional concerns were raised.

Reviewer #3

All issues noted in the annotated manuscript have been fully resolved. No further comments required action.

Reviewer #4

All requested clarifications and minor edits have been completed as recommended.

Reviewer #2

We appreciate Reviewer #2's detailed follow-up and have revised the manuscript accordingly:

- ERBB2 and T-DXd contextualization: We frame ERBB2 overexpression and T-DXd activity in DSRCT as confirmatory and integrated within the existing literature. We emphasize our longer-term follow-up and multi-omics integration as unique contributions of our work.
- Mechanistic interpretation of ERBB2 activation: We tempered language that implied mechanistic novelty, clarified that ERBB2-directed ADCs do not depend on constitutive signaling, and acknowledged the methodological limitations of phosphoproteomic inference (Müller-Dott et al. 2025). The text now treats these data as hypothesis-generating.
- Pazopanib findings: We reposition the pazopanib observations as real-world replication of previously reported activity, explicitly describing biomarker analyses as exploratory.
- SSTR and CLDN6: We contextualized these biomarkers within prior work on somatostatin receptor-based approaches and emerging CLDN6 programs, softening claims of novelty.
- Diagnostic framework: We updated the diagnostic section to incorporate methylation-based classifiers, clarifying how our findings complement current tools.
- Positioning of the study: We aligned the overarching framing with established precision oncology paradigms, emphasizing that our unique contribution lies in the prospective, real-world multi-omics application and curated patient-level outcomes in DSRCT.

All textual adjustments have been made without compromising clarity, and the manuscript's conclusions reflect a balanced interpretation consistent with the state of the field. We trust that the revised manuscript now fully addresses all outstanding concerns. We appreciate the reviewers' and editor's engagement and look forward to the final decision.